# TRACTABILITY VIA LOW DIMENSIONALITY: THE PARAMETERIZED COMPLEXITY OF TRAINING QUANTIZED NEURAL NETWORKS

**Robert Ganian**
Algorithms and Complexity Group, TU Wien, Vienna, Austria
`rganian@gmail.com`

**Frank Sommer**
Institute of Computer Science, Friedrich Schiller University Jena, Germany
`frank.sommer@uni-jena.de`

**Manuel Sorge**
Algorithms and Complexity Group, TU Wien, Vienna, Austria
`manuel.sorge@ac.tuwien.ac.at`

## ABSTRACT

The training of neural networks has been extensively studied from both algorithmic and complexity-theoretic perspectives, yet recent results in this direction almost exclusively concern real-valued networks. In contrast, advances in machine learning practice highlight the benefits of *quantization*, where network parameters and data are restricted to finite integer domains, yielding significant improvements in speed and energy efficiency. Motivated by this gap, we initiate a systematic complexity-theoretic study of ReLU Neural Network Training in the full quantization mode. We establish strong lower bounds by showing that hardness already arises in the binary setting and under highly restrictive structural assumptions on the architecture, thereby excluding parameterized tractability for natural measures such as depth and width. On the positive side, we identify nontrivial fixed-parameter tractable cases when parameterizing by input dimensionality in combination with width and either output dimensionality or error bound, and further strengthen these results by replacing width with the more general treewidth.

## 1 INTRODUCTION

A crucial task tied to the use of neural networks is their training. On a high level, this training task can be characterized as follows: given a neural network architecture $G$ and a data set $\mathcal{D}$ of input-output pairs, compute weights and biases of $G$ which minimize the error achieved by the network on $\mathcal{D}$. While we have powerful heuristics for solving this problem (Sze et al., 2017; Li et al., 2022), it also exhibits highly interesting behavior on the complexity-theoretical level and has been studied from this perspective in a series of recent foundational papers (Dey et al., 2020; Abrahamsen et al., 2021; Goel et al., 2021; Boob et al., 2022; Froese & Hertrich, 2023; Bertschinger et al., 2023; Brand et al., 2023). A detailed discussion of the state of the art is deferred to the end of this section; nevertheless, it will be useful to note that for a crisper complexity analysis one typically considers the equivalent *decision* formulation of the problem—i.e., where the input also includes an error bound $\ell$ and the algorithm is allowed to output "no" if such an error bound cannot be achieved by any combination of weights and biases.[1]

---

[1]Technically, in decision problems one is not required to output the weights and biases for positive instances; however, every algorithm obtained or mentioned in this article is constructive and capable of doing so. We note that the optimization task can be reduced to the decision formulation via a trivial search routine on $\ell$.

A common feature of all the above-mentioned complexity-theoretical works targeting the above NEURAL NETWORK TRAINING (NNT) problem is that they assume the numbers occurring in the network to be reals. This is a natural perspective that matches the classical formalization of neural networks. However, a series of recent advances have shown that one can significantly improve speed and energy efficiency by *quantizing* the neural network, i.e., forcing the numbers to lie in a specified domain of integers (Kilic et al., 2022). For example, Wang et al. (2025) recently showed that one can achieve accuracy results comparable to the real-valued setting when quantizing to 4 bits, i.e., with a domain size of 16; see also the preceding works of Yang et al. (2020) and Lin et al. (2022). Other works have also considered even stronger degrees of quantization, such as using binary domains (Lin et al., 2017; Zhu et al., 2019; Liu et al., 2020). In fact, several different methods have been developed to obtain high-quality quantized neural networks such as fully-quantized training (Zhou et al., 2016), mixed-precision training (Micikevicius et al., 2018), post-training quantization (Banner et al., 2019), and quantization-aware training (Jacob et al., 2018).

Yet, the recent developments outlined above are not at all reflected in our understanding of the underlying foundational problem: neither the complexity-theoretic lower bounds (Dey et al., 2020; Abrahamsen et al., 2021; Goel et al., 2021; Froese & Hertrich, 2023; Bertschinger et al., 2023), nor the algorithms underpinning our upper bounds for solving the training problem (Arora et al., 2018; Boob et al., 2022; Brand et al., 2023) can be translated into the quantized setting. We note that this does not seem to be merely the case of a missing "bridge" that would allow one to translate knowledge from one setting to the other—the training problem in the real-valued setting is $\exists\mathbb{R}$-complete (Abrahamsen et al., 2021; Bertschinger et al., 2023) but with quantization it is easily seen to lie in NP (see Section 2), pointing to a fundamental difference between the two settings. Until now, we lacked any complexity-theoretic study targeting NNT in the fully quantized setting.

The aim of this article is to fill the aforementioned gap by developing a comprehensive understanding of QUANTIZED RELU-NNT (see Section 2 for formal details and a discussion of the error bound):

---

**$d$-QUANTIZED RELU-ACTIVATED NEURAL NETWORK TRAINING ($d$-QNNT)**

| | |
|---|---|
| Input: | An architecture $G$ with $\alpha$ input and $\omega$ output nodes, a multiset $\mathcal{D}$ of $d$-quantized data points, and an error bound $\ell$. |
| Output: | A $d$-quantized neural network $\bar{G}$ over $G$ such that the error of $\mathcal{D}$ on $\bar{G}$ is at most $\ell$, or a correct conclusion that no such network exists. |

---

We remark that here we focus on the ReLU activation function, as it is widely used in practice and has been the target of almost all foundational studies of non-quantized NNT to date (Dey et al., 2020; Abrahamsen et al., 2021; Goel et al., 2021; Boob et al., 2022; Froese & Hertrich, 2023; Bertschinger et al., 2023; Brand et al., 2023). Our results include not only lower bounds, but also the identification of tractable cases via the development of theoretical algorithms. All our lower bounds apply already to the simplest binary quantization, while our tractability results hold for arbitrary choices of the quantization constant $d$.

In order to construct a more detailed complexity map of $d$-QNNT, we perform our analysis also taking into account the *parameterized complexity* paradigm (Cygan et al., 2015; Downey & Fellows, 2013) which associates problem instances with a suitably defined parameter, i.e., a numerical measure that captures various aspects of the instance. In the classical perspective, one would typically ask whether restricting the parameter $k$ to a constant allows us to solve instances in time polynomial w.r.t. the input size $n$. By contrast, the most desirable notion of tractability in the more refined parameterized paradigm is *fixed-parameter tractability* (FPT), meaning that the problem can be solved in time $f(k) \cdot n^{\mathcal{O}(1)}$ for some computable function $f$. To exclude inclusion in FPT, one can either show that the problem is W[1]- or W[2]-hard (which still allows for the existence of algorithms running in time, e.g., $n^{\mathcal{O}(k)}$), or NP-hard for a fixed value of $k$.

**Contributions.** For convenience, Figure 1 provides a mindmap of results that is intended to complement the description of our contributions.

Well-studied properties of the architecture $G$ that might, at first glance, seem as natural choices for parameters are its *depth* (the number of hidden layers) and *width* (the size of the largest hidden layer)—a direction which we explore in our **first set of contributions**.

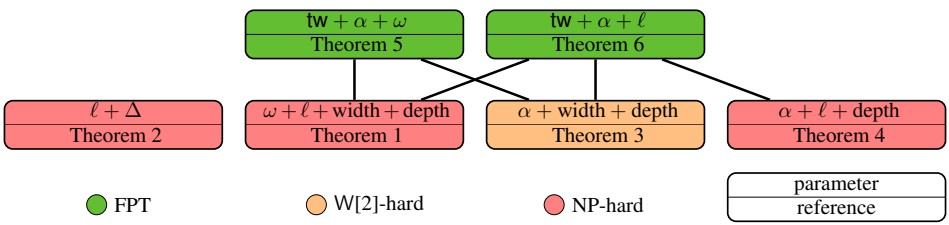

Figure 1: Overview of our results for $d$-QNNT. A combined parameter $p$ consisting of single parameters $p_1, p_2, p_3$ has an edge to a lower combined parameter $q$ if dropping one of the single parameters $p_i$ yields hardness. We use $\Delta$ to denote the maximum degree of any neuron. Our main open question concerns the complexity w.r.t. $\alpha + \omega$—see the Technical Overview and Section 5.

As a baseline result, we exclude any notion of parameterized tractability w.r.t. these two measures even when combined with the error bound $\ell$ and the output dimensionality $\omega$. In particular, in Theorem 1 we show that 2-QNNT remains NP-hard even when restricted to instances where $\ell = 0$, there is only a single output node and no hidden layer—a result which shows that even training very simple quantized architectures is computationally intractable and forms a counterpart to the well-known intractability of training a single neuron in the non-quantized setting (Goel et al., 2021; Dey et al., 2020). Naturally, the reduction underlying Theorem 1 relies on the single output neuron having large indegree—however, in our second Theorem 2 we establish the NP-hardness of 2-QNNT even on constant-degree architectures with a single hidden layer and $\ell = 0$. This latter result can be seen as a constant-degree counterpart to the $\exists$R-hardness of training shallow non-quantized networks to optimality (Abrahamsen et al., 2021).

While the above lower bounds paint a negative picture of the complexity of $d$-QNNT, there is a silver lining: both reductions inherently require the input dimensionality $\alpha$ to be large. As our **second set of contributions**, we show that parameterizing by $\alpha$ enables fixed-parameter neural network training in the quantized setting—but only when combined with additional restrictions. In particular, our results imply that for every fixed $d$, $d$-QNNT is fixed-parameter tractable w.r.t. the combined parameterizations:

1. input dimensionality $\alpha$, the width of $G$ and output dimensionality $\omega$ (Corollary 2);
2. input dimensionality $\alpha$, the width of $G$ and the error bound $\ell$ (Corollary 1).

The above results naturally lead to the question of whether all of the parameters are required to achieve fixed-parameter tractability—in other words, could any of the parameters be dropped from the statement? For $\alpha$, we already know that this is not the case: Theorem 1 rules out polynomial-time algorithms even if the width, $\omega$ and $\ell$ are small constants.

Given the fact that both positive results rely on parameterizing by the width and $\alpha$, it would be tempting to think that $d$-QNNT is fixed-parameter tractable w.r.t. $\alpha$ and the width alone—i.e., that the third parameter can be dropped in both statements. As our **third contribution**, in Theorem 3 we rule this out by establishing the W[2]-hardness of 2-QNNT w.r.t. $\alpha$ even on networks with no hidden layer. This means that neither $\omega$, nor $\ell$ can be dropped from our algorithmic upper bounds.

The above considerations leave the width as the only possible "weak point" in Corollaries 1 and 2. As our **fourth contribution**, we show that—at least if one wishes to preserve both positive results— it is neither possible to drop the width, nor replace it with the depth of $G$. In particular, our Theorem 4 shows that 2-QNNT is NP-hard even when $\alpha = 2$, there is a single hidden layer and $\ell = 0$.

While the width cannot be dropped or replaced by depth, as our **final fifth contribution** we show that Corollaries 1 and 2 can be strengthened: in particular, we prove that the results hold even if one replaces the width of architecture $G$ with its *treewidth* $\mathsf{tw}(G)$ (Robertson & Seymour, 1984). The latter is a well-established measure of the tree-likeness of a graph; on architectures with hidden neurons it never exceeds the width, but can be arbitrarily smaller. For example, an architecture consisting of layers whose width alternates between small and large will have large width, but small treewidth. Thus, while non-trivial to prove, the following two results supersede and directly imply Corollaries 1 and 2:

1*. $d$-QNNT is fixed-parameter tractable w.r.t. $\alpha + \mathsf{tw}(G) + \omega$ (Theorem 5);

$2^\star$. $d$-QNNT is fixed-parameter tractable w.r.t. $\alpha + \mathsf{tw}(G) + \ell$ (Theorem 6).

**Technical Overview.** To obtain our lower bounds, we develop targeted reductions from a variety of problems, including BOOLEAN SATISFIABILITY, HITTING SET, and SET COVER. While each of the reductions is distinct, the constructed architectures are often very dense and have simple graph structures. In other words, our results show that the difficulty of training in the quantized setting does not stem from the complexity of the architecture, but rather from the presence of high-dimensional data on the input or output. In fact, the main open question arising from our work is whether the converse is true: can we efficiently solve instances of $d$-QNNT with possibly complicated architectures, but constant input and output data dimensionality (i.e., $\alpha + \omega$)?

For our positive results—specifically, Theorems 5 and 6—the main technical difficulty is that the trained $n$-node networks could contain hidden neurons with $\Theta(n)$ incoming arcs from the preceding layer that have non-zero weights. Indeed, it is not difficult to construct instances with such solutions—and yet the dynamic programming techniques that form the cornerstone of most treewidth-based algorithms are incapable of efficiently searching for them. To deal with this issue, we make a detour and first establish a structural insight that we believe is of independent interest: every YES-instance of $d$-QNNT admits at least one solution where the number of activated arcs entering any node is upper-bounded by a function of the parameters. This is formalized in Lemma 1, and relies on an involved proof that builds on Steinitz' Lemma.

*Full proofs and details deferred to the Full version, which can be found in the supplemental material, are marked with ($\star$).*

**Related Work.** Beyond the related articles mentioned in the second paragraph, several of the earlier works in the field also studied (the complexity of) NNT in the partially quantized setting (Judd, 1988; Blum & Rivest, 1992; Parberry, 1992; Courbariaux et al., 2015; Zhu et al., 2017) or with different activation functions (Judd, 1990; Schmitt, 2004; Doron-Arad, 2025). In particular, the NP-hardness of 2-QNNT can be inferred from the reduction in the seminal work of Judd (1990, Theorem 24) on training Boolean neural networks with AND and OR gates, and separately also from the reduction in Schmitt (2004, Theorem 7) using linear threshold activation functions. However, our Theorems 1 to 4 obtain lower bounds in conjunction with additional restrictions on the inputs that are required for our parameterized lower bounds. Crucially, we are aware of neither any in-depth multivariate complexity analysis in this setting, nor any works directly targeting the complexity of quantized neural network training with ReLU activation functions. ($\star$)

## 2  PRELIMINARIES

For an integer $d \geq 1$, we define the *$d$-quantized integer domain* $\mathbb{Z}_d$ as $\{z \in \mathbb{Z} \mid -\lfloor \frac{d-1}{2} \rfloor \leq z \leq \lceil \frac{d-1}{2} \rceil\}$, that is, $\mathbb{Z}_2 = \{0, 1\}$, $\mathbb{Z}_3 = \{-1, 0, 1\}$, $\mathbb{Z}_4 = \{-1, 0, 1, 2\}$ and so forth[2]. The *$d$-domain ReLU activation function* $\mathrm{ReLU}_d : \mathbb{Z}_d \to \mathbb{Z}_d$ is the restriction of the well-known rectified linear unit to $\mathbb{Z}_d$—that is, all negative values are mapped to 0 while on positive values $\mathrm{ReLU}_d$ is the identity except that inputs outside of $\mathbb{Z}_d$ become $\max \mathbb{Z}_d$.

We say that a *network architecture* is a directed acyclic graph (a *DAG*) $G$ whose vertex sets are partitioned into *layers*, where layer 0 consists solely of sources, and such that an arc $ab$ may only go from a vertex in layer $i$ (for $i \in \mathbb{N}$) to a vertex in layer $i + 1$ and all sinks lie in the same layer. We will refer to the *sources* and *sinks* the *input* and *output* neurons of $G$, respectively, while all other nodes of $G$ are referred to as *hidden neurons*. We assume that the sources are equipped with a fixed ordering, and the same also for the sinks. The maximum size of a layer with only hidden neurons is called the *width* of $G$, while we refer to the number of layers as the *depth* of $G$.

Let us fix a $d$-quantized integer domain $\mathbb{Z}_d$. A neural network $\bar{G}$ over an architecture $G$ is a tuple $(G, \texttt{weight}, \texttt{bias})$ where the weight function $\texttt{weight}$ assigns each arc of $G$ a weight from $\mathbb{Z}_d$, and the bias function $\texttt{bias}$ assigns each non-source node of $G$ a bias from $\mathbb{Z}_d$. Let the number

---

[2]Our model matches, e.g., the so-called "E1M2" format of the 4-bit floating point standard FP4. Other low-bit number encodings have also been considered in the quantized setting (Wang et al., 2025), but we focus our exposition on this theoretically cleanest model. While we do not formally prove this, all obtained results seem to readily carry over to different low-bit number encodings with only minor modifications to the proofs.

of input and output neurons of $G$ be $\alpha$ and $\omega$, respectively. The *evaluation* of an input data vector $\vec{x} \in (\mathbb{Z}_d)^\alpha$ is a mapping $f$ which assigns each node of $G$ a *value* (or *activation*) computed as follows:

- The $i$-th input neuron receives the value $\vec{x}[i]$;
- For each neuron $v \in V(G)$ with predecessors $z_1, \ldots, z_q$, we set its value as[3] $\text{ReLU}_d\big((\sum_{i \in [q]} f(z_i) \cdot \texttt{weight}(z_i v)) - \texttt{bias}(v)\big)$.

The input to $\text{ReLU}_d$ above is sometimes called the *pre-activation value*. Given a data point $p \in \mathcal{D}$, we say that a neuron $q$ is *active* in $\bar{G}$ if in the evaluation of $p$, the neuron $q$ receives a positive activation; otherwise, it is *inactive*. We denote the restriction of $f$ to the output nodes, represented as a vector of integers in $(\mathbb{Z}_d)^\omega$ ordered by the output neurons, as the *output* of the neural network on $\vec{x}$. In the training setting, we will be dealing with $d$-quantized data points from $(\mathbb{Z}_d)^\alpha \times (\mathbb{Z}_d)^\omega$. The *error* of a multiset of such data points is equal to the number of misaligned data points, i.e., the number of pairs $(\vec{x}, \vec{y})$ in the multiset such that the output of $(G, \texttt{weight}, \texttt{bias})$ on $\vec{x}$ differs from $\vec{y}$. With these definitions in place, we study $d$-QNNT as formalized in Section 1.

$d$-QNNT is in NP (a certificate consists of a linear number of integers from $\mathbb{Z}_d$), which contrasts the $\exists\mathbb{R}$-completeness of the training problem in the non-quantized setting. In the non-quantized setting, one typically uses a wide variety of loss functions tailored to real-valued errors such as $\ell_2^2$ (Brand et al., 2023)—here, we focus on a simple error count (as also used, e.g., by Judd (1990)) in order to facilitate a cleaner analysis. The majority of our proofs could nevertheless be directly and straightforwardly translated to other loss functions (this is easiest to see for Theorems 1, 2, 4, 5).

**Treewidth.** A *tree decomposition* $\mathcal{T}$ of an undirected graph $G$ (or the underlying undirected graph of a directed graph) is a pair $(T, \chi)$, where $T$ is a tree and $\chi$ is a function that assigns each tree node $t$ a set $\chi(t) \subseteq V(G)$ of vertices such that the following conditions hold: **(P1)** for every edge $e \in E(G)$ there is a tree node $t$ such that $e \subseteq \chi(t)$; and **(P2)** for every vertex $v \in V(G)$, the set of tree nodes $t$ with $v \in \chi(t)$ induces a non-empty subtree of $T$. The sets $\chi(t)$ are called *bags* of the decomposition $\mathcal{T}$, and $\chi(t)$ is the bag associated with the tree node $t$. The *width* of a tree decomposition $(T, \chi)$ is the size of a largest bag minus 1. The *treewidth* of a graph $G$, denoted by $\text{tw}(G)$, is the minimum width over all tree decompositions of $G$.

*A detailed treatment of parameterized complexity and treewidth is provided in the appendix ($\star$).*

## 3  Lower Bounds for $d$-QNNT

In this section, we show that 2-QNNT remains intractable in highly restrictive settings. First, in Theorem 1, we establish NP-hardness even if the architecture has no hidden neuron, only one output neuron, and for training without error. Note that Theorem 1 implies NP-hardness even when the combined parameter width + depth + $\ell$ + $\omega$ is upper-bounded by a constant. Naturally, the corresponding reduction requires the output neurons to have an arbitrarily large degree. One could hence hope that architectures with constant maximum degree can be trained efficiently. In Theorem 2, we show that this is not possible by establishing NP-hardness for this setting.

In both the reductions that underlie Theorems 1 and 2 the number of input neurons is large and in particular not upper-bounded by a function of the parameters. Hence, one could hope that a small or even constant number of inputs allows for efficient training. We show that this is not the case either. First, in Theorem 3, we provide W[2]-hardness for $\alpha$ even if there is no hidden layer. Second, in Theorem 4, we show that 2-QNNT remains NP-hard even if there are only 2 inputs and 1 hidden layer. Altogether, these results yield the lower bounds depicted in Figure 1.

**Theorem 1** ($\star$)**.** 2-QNNT *is* NP-*hard even when restricted to instances where $\ell = 0$ and architectures with a single output neuron and no hidden neuron.*

*Proof Sketch.* We provide a reduction from the NP-hard Exact Set Cover problem (Karp, 1972) where the input consists of a universe $U$, and a family $\mathcal{F}$ of subsets over $U$. The goal is to find a subset $\mathcal{S} \subseteq \mathcal{F}$ such that $\mathcal{S}$ is a partition of $U$, that is, 1) $\bigcup_{S \in \mathcal{S}} S = U$ and 2) $S_1 \cap S_2 = \emptyset$ for each $S_1, S_2 \in \mathcal{S}$.

---

[3]We note that the bias is subtracted instead of added to the result due to the fact that, in the Boolean-domain case, subtracting allows the bias to actually interact with the weights (see also Kilic et al. (2022)). For larger domains, the distinction is inconsequential since we can flip the sign of the bias.

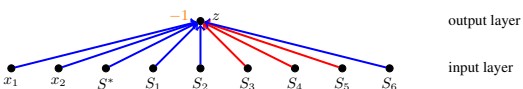

Figure 2: An illustration of the reduction behind Theorem 1 for the universe $U = [6]$ and the set family $\mathcal{F}$ with sets $S_1 = \{1, 4, 5\}, S_2 = \{2, 3\}, S_3 = \{1, 6\}, S_4 = \{2, 5\}, S_5 = \{3, 5\}, S_6 = \{6\}$ with an exact set cover $\mathcal{S} = \{S_1, S_2, S_6\}$. In the solution corresponding to $\mathcal{S}$, each red arc has weight 0 and each blue arc has weight 1. The orange number is the bias of the output neuron.

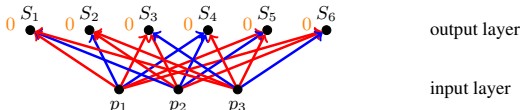

Figure 3: An illustration of the reduction behind Theorem 3 for the universe $U = [6]$ and the set family $\mathcal{F}$ with sets $S_1 = \{1, 4, 5\}, S_2 = \{2, 3\}, S_3 = \{1, 6\}, S_4 = \{2, 5\}, S_5 = \{3, 5\}, S_6 = \{6\}$ and $k = 3$ and with a hitting set $S = \{2, 5, 6\}$. In the solution corresponding to $S$, inputs $p_1, p_2$ and $p_3$ are associated with elements 2, 5 and 6, respectively. Moreover, each red arc has weight 0 and each blue arc has weight 1. The orange numbers are the biases of the output neurons.

We construct an equivalent instance $I$ of 2-QNNT as follows; see Figure 2 for an illustration.
*Description of the architecture $G$.* Abusing notation, for each set $F \in \mathcal{F}$ we create a *set input neuron $F$*. Moreover, we add 3 more *dummy input neurons $S^*$, $x_1$, and $x_2$*, respectively. Finally, we add one output neuron $z$ and add an arc from each input neuron to the unique output neuron $z$.
*Description of the data set.* For each element $u \in U$ we add two *element data points: $d_u^1$ and $d_u^2$*: both have value 1 in each input corresponding to a set containing $u$ and value 0 in dummy inputs $x_1$ and $x_2$. Moreover, $d_u^1$ has value 0 in dummy input $S^*$ and value 0 in output $z$, and $d_u^2$ has value 1 in dummy input $S^*$ and value 1 in output $z$. Finally, we add three further data points: *dummy data points $d_{01}, d_{10}$, and $d_{11}$*. All three have value 0 in each set input and in dummy input $S^*$. Moreover, $d_{01}$ has values $x_1 = 0$, $x_2 = 1$ and output value 0, $d_{10}$ has values $x_1 = 1$, $x_2 = 0$ and output value 0, and $d_{11}$ has values $x_1 = 1$, $x_2 = 1$ and output value 1.

Finally, we set $\ell = 0$. To complete the proof, it remains to establish correctness. $(\star)$ $\qquad\square$

We note that one could also obtain Theorem 1 by carefully adapting the hardness proof of Schmitt (2004, Theorem 7) to our setting. However, the reduction we provide here is simpler, self-contained, and additionally also implies W[1]-hardness with respect to the number of arcs with weight one in the solution. We continue by stating the hardness for constant-degree architectures; since this result is not central to our complexity landscape (see Figure 1), we defer its proof to the appendix.

**Theorem 2** $(\star)$. *2-QNNT is NP-hard even when restricted to instances where $\ell = 0$, $|\mathcal{D}| \leq 4$, and architectures with only one hidden layer, maximum outdegree 3, and maximum indegree 2.*

Next, we establish W-hardness w.r.t. the number $\alpha$ of inputs even if there is no hidden layer.

**Theorem 3** $(\star)$. *Even if the network has no hidden neuron, 2-QNNT is W[2]-hard when parameterized by the number $\alpha$ of input nodes, even when restricted to architectures with no hidden neurons.*

*Proof Sketch.* We present a reduction from the HITTING SET (HS) problem where the input consists of a universe $U$, a family $\mathcal{F}$ of subsets over $U$, and an integer $k$. The goal is to find a subset $S \subseteq U$ (called a *hitting set*) of size $k$ such that $S$ contains at least one element of each set in the family, that is, $S \cap F \neq \emptyset$ for any $F \in \mathcal{F}$. HS is W[2]-hard parameterized by $k$ (Cygan et al., 2015).

We construct an instance $I$ of 2-QNNT as follows. For an illustration, see Figure 3.
*Description of the architecture $G$.* We create $k$ input neurons $p_1, \ldots, p_k$. Abusing notation, for each set $F \in \mathcal{F}$ we create one *set output neuron $F$*. We add arcs between every input and output neuron.
*Description of the data set.* For each element $u \in U$ we add $k$ *element $u$ data points $d_u^1, \ldots, d_u^k$*. Element $u$ data point $d_u^i$ has value 1 in input $p_i$ and value 0 in each other input. Moreover, $d_u^i$ has value 1 in each set output $F$ such that $u \in F$. Thus, $d_u^i$ has value 0 in each set output $F'$ such that $u \notin F'$. Observe that the $k$ element $u$ data points all have the same output but they have pairwise

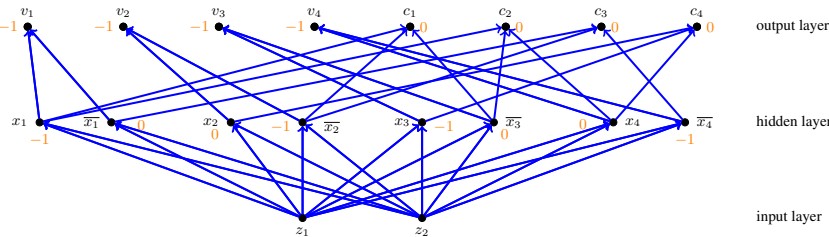

Figure 4: An illustration of the reduction behind Theorem 4 for the formula $\Phi$ with clauses $c_1 = x_1 \vee \overline{x_2} \vee \overline{x_3}$, $c_2 = x_1 \vee \overline{x_3} \vee x_4$, $c_3 = \overline{x_1} \vee \overline{x_2} \vee \overline{x_4}$, and $c_4 = x_2 \vee x_3 \vee x_4$ with a satisfying assignment $\mathcal{A}$ with $\{x_2, x_4\} \mapsto \texttt{true}$ and $\{x_1, x_3\} \mapsto \texttt{false}$. In an optimal solution all arcs have weight 1. The biases of of a solution corresponding to $\mathcal{A}$ are shown in orange.

different inputs. Then, we add a *verifier data point* $d^*$ which has value 1 in each input and in each output. In the following, we say that two data points $d_1$ and $d_2$ have the same *type* if the input values of $d_1$ and $d_2$ are pairwise identical. Note that we have exactly $k + 1$ distinct types of data points.

Finally, we set $\ell := k \cdot (|U| - 1)$. To complete the proof, it remains to establish correctness. $(\star)$ $\quad\square$

For our fourth lower bound, we use a "compressed" version of the construction behind Theorem 2 to obtain NP-hardness for only 2 input nodes and 3 data points.

**Theorem 4** $(\star)$. *2-QNNT is* NP-*hard even if* $\alpha = 2$, $\ell = 0$, $|\mathcal{D}| = 3$, *and depth* $= 1$.

*Proof Sketch.* We present a reduction from 3-SAT (Karp, 1972), where one is given a CNF formula $\Phi$ on variables $x_1, \ldots, x_n$ and a set of $m$ clauses each consisting of precisely three literals.

We construct an equivalent instance $I$ of 2-QNNT as follows; see Figure 4 for an illustration.
*Description of architecture $G$.* We create two input neurons $z_1$ and $z_2$. For each of the two literals of a variable $x_i$ with $i \in [n]$, we create two *hidden neurons* $x_i$ and $\overline{x_i}$ *associated with variable* $x_i$. Thus, we create $2n$ hidden neurons. Moreover, we create a *variable output neuron* $v_i$ *associated with variable* $x_i$ for each variable $x_i$. Also, we add one *clause output neuron* $c_j$ for each clause of $\Phi$. Thus, we create $n+m$ output neurons. We add an arc from each input neuron to each hidden neuron. Next, we add an arc from each of the two hidden neurons $x_i$ and $\overline{x_i}$ associated with variable $x_i$ to the variable output neuron $v_i$ associated with variable $x_i$. Finally, for each clause $c_j$ consisting of literals $p_1, p_2$, and $p_3$, we add the arcs $(p_h, c_j)$ for each $h \in [3]$.
*Description of data set.* Here, we use the notation $(z_1, z_2) \mapsto (V, C)$ for the data points, where $z_1$ and $z_2$ are numbers referring to the inputs, and $V$ and $C$ are vectors referring to the outputs. More precisely, $V$ has length $n$, and the $i$-th entry corresponds to the variable output neuron $v_i$, and $C$ has length $m$, and the $j$-th entry corresponds to the clause output neuron $c_j$. Whenever we put a 0 or a 1 in any of the three vectors, we mean that all corresponding outputs receive value 0 or 1, respectively.

We add 3 data points: **(1)** The *verifier 1 data point* with $(1, 0) \mapsto (0, 1)$, **(2)** the *verifier 2 data point* with $(0, 1) \mapsto (0, 1)$, and **(3)** the *choice data point* with $(1, 1) \mapsto (1, 1)$. Finally, we set $\ell := 0$.

**Intuition.** Recall that we say that given a data point $p$ a neuron $q$ is *active* if in the evaluation of $p$, the neuron $q$ receives a positive activation; otherwise, it is *inactive*. The idea is that when considering the verifier 1 data point, the active hidden neurons correspond to a satisfying variable assignment. We achieve this with the variable output neurons: If both hidden neurons $x_i$ and $\overline{x_i}$ associated with a variable $x_i$ are active for the verifier 1 data point, then since the value of the variable output neuron $v_i$ associated with $x_i$ needs to be 0 and since $x_i$ and $\overline{x_i}$ are the unique neighbors of $v_i$ this then implies that the value of $v_i$ for the choice data point is also 0, and not 1 as desired, yielding an error.

To complete the proof, it remains to use the above intuition to formally establish correctness. $(\star)$ $\quad\square$

## 4 FIXED-PARAMETER TRACTABILITY

In this section we prove our tractability results for parameter combinations that include the width, treewidth, and number $\alpha$ of input neurons. We begin by showing a structural result (Lemma 1)

that states that there is always a solution that has upper-bounded degree in the sense that, for each neuron, there is only a bounded number of incoming arcs with nonzero weights. We then use Lemma 1 to prove tractability of $d$-QUANTIZED RELU-ACTIVATED NEURAL NETWORK TRAINING ($d$-QNNT) without error with respect to the treewidth and number $\alpha$ of input neurons (Lemma 3). Then we show how to lift this result to training with nonzero error bounds and how the treewidth results imply the corresponding results for the width.

Consider a neuron $v$ in a neural network. Define the *non-zero in-neighbors* of $v$ to be the in-neighbors $u$ of $v$ such that `weight`$(uv) \neq 0$. The *non-zero indegree* of $v$ is the number of non-zero in-neighbors.

**Lemma 1** ($\star$). *Let $G$ be an architecture and $\mathcal{D}$ a data set with $p$ distinct input vectors. If there is a neural network over $G$ with zero error on $\mathcal{D}$, then there is a neural network $\bar{G}$ over $G$ with zero error on $\mathcal{D}$ such that for each neuron $v$ in $\bar{G}$ the number of non-zero in-neighbors of $v$ is at most $(dp)^{\mathcal{O}(p)}$.*

We prove Lemma 1 by using Steinitz' Lemma, stated as follows.

**Lemma 2** (Steinitz' Lemma (Steinitz, 1913; Sevast'janov, 1994)). *Let $\|\cdot\|$ be an arbitrary norm on $\mathbb{R}^d$. Let $x_1, \ldots, x_m \in \mathbb{R}^d$ such that $\sum_{i \in [m]} x_i = 0$ and for each $i \in [m]$ we have $\|x_i\| \leq 1$. Then there exists a permutation $\pi \in S_m$ such that all prefix sums have norm at most $d$. That is, for each $k \in [m]$ we have $\|\sum_{j \in [k]} x_{\pi(j)}\| \leq d$.*

*Proof Sketch for Lemma 1.* Consider a neuron $v$ in a solution network. We can collect the activations of $v$ for each input vector in a vector $\vec{s} \in (\mathbb{Z}_d)^p$. Assume for simplicity that we don't have ReLU activations and instead simply pass through the weighted sum of the activations of the in-neighbors and, furthermore, each of the summed activations is in $(\mathbb{Z}_d)^p$. Then, $\vec{s}$ is a small-norm vector and it is obtained as a sum of small-norm vectors. Steinitz' Lemma tells us that we can reorder the vectors such that each prefix sum has small norm. This means that, if there are many non-zero in-neighbors to $v$, then at least one prefix sum occurs twice. This means that the vectors in between these two identical sums sum to zero and we can simply set their corresponding arc weights to zero without changing the activation of $v$. Care must be taken to preserve the ReLU activations and boundaries of $(\mathbb{Z}_d)^p$ and to ensure that all vectors in the sum have small norm. $\square$

We next show how the degree bound above can be used to efficiently train neural networks for low-treewidth architectures and small number of input neurons. We will use a dynamic program over a tree decomposition. Essentially this means that we need to maintain for small separators what the status of partial solutions on one side, say the left side, of the separator is and this status needs to be encoded in a small number of states. Consider a neuron $v$ in such a separator. We want to maintain as a state of the partial solution which pre-activation values $v$ has already received on the left side of the separator. If the non-zero indegree of a solution is large, then we may have already seen an unbounded number of negative pre-activation values, but on the right side we may still see an equally large number of positive pre-activation values, in total summing to a small value in $\mathbb{Z}_d$. To properly maintain the activation of $v$, we would thus need to maintain unboundedly large pre-activation values, leading to a large, unbounded number of dynamic-programming states. In contrast, using the indegree bound established in Lemma 1, we can assume that the sums of pre-activation values are bounded and only look for such solutions.

**Lemma 3** ($\star$). *$d$-QNNT with $\ell = 0$ is* FPT *w.r.t. the treewidth of $G$ and the number of input nodes.*

*Proof Sketch.* Let $(G, \alpha, \omega, d, \mathcal{D}, 0)$ be an instance of $d$-QNNT with error bound $\ell = 0$ and $\alpha$ input nodes (i.e., neurons). Let $\mathcal{X}$ be the set of distinct input vectors in $\mathcal{D}$ and tw be the treewidth of the input architecture $G$. First, we compute a tree decomposition $\mathcal{T} = (T, \chi)$ of the underlying undirected graph of the architecture $G$ that has width at most $2\text{tw} + 1$ (Korhonen, 2022). We then proceed by dynamic programming on $\mathcal{T}$. Without loss of generality, there are at most $d^\alpha$ different input vectors (otherwise either there are multiple pairs of equal pairs of input and output vectors, of which we can drop one arbitrarily, or one input vector is associated with two different output vectors, and we have a trivial no-instance). Thus, by Lemma 1 we know that, if there is a solution neural network, then there is a solution with non-zero indegree at most $(d(d^\alpha))^{\mathcal{O}(d^\alpha)} = d^{\mathcal{O}(\alpha d^\alpha)}$. We hence try to find a solution with non-zero indegree at most some integer $\Delta := d^{\mathcal{O}(\alpha d^\alpha)}$. (Indeed, we won't enforce this indegree bound, but we are guaranteed to find a solution, potentially with larger non-zero indegree, if there is one.)

*Partial neural networks and evaluations thereof.* To define the dynamic-programming table, we need to define what a partial solution is for the part of the architecture we have already seen in the dynamic program. Let $W \subseteq V(G)$. A *W-partial* neural network over architecture $G$ is a tuple $(G, \texttt{weight}, \texttt{bias})$, where $\texttt{weight}$ and $\texttt{bias}$ are defined in the same way as for neural networks except that the domain of $\texttt{bias}$ is $W$ and the domain of $\texttt{weight}$ is the set of arcs of $G$ with both endpoints in $W$. Note that the activation value for a neuron $v$ on a certain input vector is defined if for each path $P$ in $G$ from an input neuron to $v$ all biases and weights of neurons and arcs on $P$ are defined. Below we will additionally refer to activation values for further neurons based on assuming that they receive certain given weighted activation values from in-neighbors where biases or weights are not defined. More precisely, for a $W$-partial neural network, consider an input vector $x$. For some neurons $v$, including all of those whose in-neighbors are not all contained in $W$, we additionally specify the weighted activation value $\texttt{future}(x, v)$ that they receive from the in-neighbors not contained in $W$. This is sufficient to compute the activation values (as defined for non-partial neural networks) for all neurons in $W$, based on assuming the values $\texttt{future}(x, v)$. Below we will omit explicit mention of this assumption when referring to the activation values as long as it is clear from the context.

*The dynamic programming table.* Below, for a node $t \in V(T)$ in the tree decomposition we define $V_t$ to be the union of all bags of nodes that are either $t$ or descendants of $t$ in $T$. The dynamic-programming table $D$ is defined as follows. (Recall that $\mathcal{X}$ is the set of input vectors.) Consider a node $t \in V(T)$ in the tree decomposition, a function $\texttt{bias} \colon \chi(t) \to \mathbb{Z}_d$ assigning a bias to each neuron in $t$'s bag, a function $\texttt{weight} \colon \{(u,v) \in E(G) \mid u, v \in \chi(t)\} \to \mathbb{Z}_d$ assigning a weight to each arc in $t$'s bag, a function $\texttt{seen} \colon \mathcal{X} \times \chi(t) \to \mathbb{Z}_{d^2\Delta}$ assigning each neuron in $t$'s bag a set of pre-activation values received from neurons in $V_t$, and a function $\texttt{future} \colon \mathcal{X} \times \chi(t) \to \mathbb{Z}_{d^2\Delta}$ assigning each neuron in $t$'s bag a set of pre-activation values to be received from neurons in $V \setminus V_t$. We put $D[t, \texttt{bias}, \texttt{weight}, \texttt{seen}, \texttt{future}] = 1$ if there is a $V_t$-partial neural network $\bar{G}$ over $G$ with the following properties, where all references to activation values are with respect to $\bar{G}$:

(i) For each neuron $v$ in $\chi(t)$ its bias in $\bar{G}$ is $\texttt{bias}(v)$, and for each arc $(u,v) \in E(G)$ with $u, v \in \chi(t)$ the arc weight in $\bar{G}$ is $\texttt{weight}(u,v)$.
(ii) For each input vector $x \in \mathcal{X}$, assuming that for each neuron $v \in \chi(t)$ the pre-activation value received from in-neighbors in $V(G) \setminus V_t$ is $\texttt{future}(x, v)$, then for each neuron $v \in \chi(t)$ the pre-activation value received from in-neighbors in $V_t$ is $\texttt{seen}(x, v)$.
(iii) For each input vector $x \in \mathcal{X}$, for each input neuron in $V_t \setminus \chi(t)$ the activation value is exactly the one specified in $x$.
(iv) For each input-output pair $(x, y)$, for each output neuron $v \in V_t \setminus \chi(t)$, the activation of $v$ on input $x$ is exactly as specified in $y$.

If there is no such neural network $\bar{G}$ then we put $D[t, \texttt{bias}, \texttt{weight}, \texttt{seen}, \texttt{future}] = 0$.

The computation of the table $D$ for each node of $T$ and the running time is in the appendix. $\qquad\square$

Instances with nonzero error bounds can be reduced to the $\ell = 0$ setting in order to apply Lemma 3.

**Theorem 5** ($\star$). *$d$-QNNT is* FPT *wrt. the treewidth of $G$, the number $\alpha$ of input dimensions, and the number $\omega$ of output dimensions.*

**Theorem 6** ($\star$). *$d$-QNNT is* FPT *w.r.t. the treewidth of $G$, the number $\alpha$ of input dimensions, and the error bound $\ell$.*

Finally, we show that the treewidth tw can be replaced by the width. If there is at least one hidden layer, then we can show that indeed the width is an upper bound for tw and Theorems 5 and 6 directly apply. Otherwise, we design two simple ad-hoc strategies that learn the neural networks optimally.

**Corollary 1** ($\star$). *$d$-QNNT is* FPT *with respect to $\alpha + \ell + width$.*

**Corollary 2** ($\star$). *$d$-QNNT is* FPT *with respect to $\alpha + \omega + width$.*

## 5    CONCLUDING REMARKS

Our work initiates the study of fully quantized ReLU neural network training from the classical as well as parameterized complexity perspectives. We show that the problem remains NP-hard even

in highly restricted settings, but also provide positive results through the identification of non-trivial fixed-parameter tractable fragments. We remark that the latter outcome contrasts the state of the art for neural network training in the non-quantized setting. Indeed, in spite of being targeted by several recent complexity-theoretic studies (Dey et al., 2020; Abrahamsen et al., 2021; Goel et al., 2021; Boob et al., 2022; Froese & Hertrich, 2023; Bertschinger et al., 2023; Brand et al., 2023), to date we do not know a single *non-trivial*[4] parameterization that yields fixed-parameter tractability for training non-quantized neural networks. Moreover, we believe that settling the parameterized complexity of $d$-QNNT w.r.t. the input and output dimensionality (i.e., $\alpha + \omega$) will require insights beyond the current state of the art and pose this as the main open question arising from our work. Other important avenues of future work include whether our results can be extended to distillation, and whether they could be used to obtain more efficient empirical algorithms.

ACKNOWLEDGMENTS

Robert Ganian acknowledges support by the Austrian Science Fund (FWF projects 10.55776/Y1329 and 10.55776/COE12).

Frank Sommer was supported by the Alexander von Humboldt Foundation and partially by the Carl Zeiss Foundation, Germany, within the project "Interactive Inference".

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
