**Related Work.** The complexity of non-quantized NEURAL NETWORK TRAINING has been studied predominantly in the ReLU-activated setting (i.e., the one targeted in our article). The only other setting considered in complexity-theoretic studies to date is the one with linearly activated neurons; there, the non-quantized problem was shown to be $\exists$R-complete (Abrahamsen et al., 2021) but polynomial-time solvable for certain special classes of architectures (Brand et al., 2023). For ReLU-activated neurons, the non-quantized training problem is known to be $\exists$R-complete even when restricted to exact training on architectures with two input neurons, two output neurons and two hidden layers (Bertschinger et al., 2023). A series of works have shown that the same training problem is computationally intractable also when restricted to architectures with a single hidden neuron (Dey et al., 2020; Goel et al., 2021; Froese et al., 2022; Froese & Hertrich, 2023). In terms of upper bounds, Arora et al. (2018) established polynomial-time tractability when training non-quantized instances with a single non-activated output neuron; their result was subsequently improved to an activated output neuron (Boob et al., 2022), and most recently generalized to architectures with maximum output degree of at most one (Brand et al., 2023).

Apart from the articles on fully-quantized neural networks mentioned in the second paragraph, we remark that several of the earlier works in the field also considered models where only the activations are quantized but not the data (Courbariaux et al., 2015; Zhu et al., 2017). Moreover, Judd (1988), Blum & Rivest (1992), and Parberry (1992) established the NP-hardness of training partially quantized networks over 30 years ago; in their models, the data/signals are quantized but not the activations. These latter results also hold for highly restricted architectures, including planar architectures (Judd, 1988) and architectures of constant internal width (Blum & Rivest, 1992).

We note that algorithms and lower bounds for training fully quantized neural networks have been studied in a handful of past works, but not for the standard ReLU activation function considered here. In his dissertation, Judd (1990) established lower bounds for Boolean NNT with activations modeled as AND and OR gates rather than ReLU. Schmitt (2004) studied fully quantized NNT with linear activations and also quantized NNT where the thresholds (i.e., biases) are not restricted by quantization. Finally, the very recent work of Doron-Arad (2025) considers quantized NNT with division-based activation functions.

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

For presenting our dynamic-programming algorithms, it is convenient to consider tree decompositions in the following normal form Kloks (1994): A tree decomposition $(T, \chi)$ is a *nice tree decomposition* of a graph $G$ if the tree $T$ is rooted at a node $r$, and each node of $T$ is of one of the following four types:

1. a *leaf node*: a node $t$ having no children and $|\chi(t)| = 1$;
2. an *introduce node*: a node $t$ having exactly one child $t'$, and $\chi(t) = \chi(t') \cup \{v\}$ for a node $v$ of $G$;
3. a *forget node*: a node $t$ having exactly one child $t'$, and $\chi(t) = \chi(t') \setminus \{v\}$ for a node $v$ of $G$;
4. a *join node*: a node $t$ having exactly two children $t_1, t_2$, and $\chi(t) = \chi(t_1) = \chi(t_2)$.

For convenience we will also assume that $\chi(r) = \emptyset$ for the root $r$ of $T$. We can achieve this straightforwardly by introducing forget nodes above the root until its bag is empty.

Given a graph $G$ with treewidth tw, a tree decomposition of width at most 2tw+1 can be computed in $2^{\mathcal{O}(\mathrm{tw})} \cdot |V(G)|$ time (Korhonen, 2022). A tree decomposition $\mathcal{T}$ of width tw can be turned into a nice tree decomposition of the same width and with $\mathcal{O}(\mathrm{tw}|V(G)|)$ nodes in $\mathcal{O}(\mathrm{tw} \cdot \max(|V(G)|, |V(T)|))$ time (Cygan et al., 2015, Lemma 7.4).

As mentioned in the introduction, our fixed-parameter algorithms that utilize treewidth (Theorems 5 and 6) generalize and imply the corresponding results for width. To see this, we prove the following structural observation:

**Observation 1.** *For each architecture $G$ containing at least one hidden neuron, $\mathrm{tw}(G)$ is upper-bounded by twice the width of $G$.*

*Proof.* Let $V_{\mathrm{in}}, V_i, V_{\mathrm{out}}$ denote the input neurons, hidden neurons in layer $i \in [q]$ where $q$ is the depth, and the output neurons, respectively. We construct a tree decomposition $\mathcal{T}$ with the desired width as follows: **(1)** For each $v_{\mathrm{in}} \in V_{\mathrm{in}}$ we create a bag consisting of $v_{\mathrm{in}} \cup V_1$ (in-bags), **(2)** for each $i \in [q-1]$ we create a bag consisting of $V_i \cup V_{i+1}$ (inner-bags), and **(3)** for each $v_{\mathrm{out}} \in V_{\mathrm{out}}$ we create a bag consisting of $v_{\mathrm{out}} \cup V_q$ (out-bags). The bags are connected as follows: **(1)** Each in-bag is adjacent to the inner-bag $V_1 \cup V_2$, **(2)** inner-bag $V_i \cup V_{i+1}$ is adjacent to the inner bag $V_{i+1} \cup V_{i+2}$, and **(3)** each out-bag is adjacent to the inner-bag $V_{q-1} \cup V_q$. The claim follows by the fact that each bag in $\mathcal{T}$ either forms a subset of two hidden layers, or is a hidden layer plus a single neuron. $\square$

On the other hand, note that $\mathrm{tw}(G)$ can be arbitrarily smaller than the width since very large hidden layers can alternate with very small hidden layers (in which case one can construct a tree decomposition whose width is twice the size of the smaller hidden layers).

**Parameterized Complexity.** In parameterized complexity (Downey & Fellows, 2013; Cygan et al., 2015), the running-times of algorithms are studied with respect to a parameter $p \in \mathbb{N}$ and input size $n$. It is normally used for NP-hard problems, with the aim of finding a parameter describing a feature of the instance such that the combinatorial explosion is confined to this parameter. A parameterized problem is *fixed-parameter tractable* (FPT) if it can be solved by an algorithm running in time $f(p) \cdot n^{\mathcal{O}(1)}$, where $f$ is a computable function

Proving that a problem is W[2]-hard (or W[1]-hard) via a *parameterized reduction* from a W[2]-hard (W12]-hard, respectively) problem $\mathcal{P}$ rules out the existence of a fixed-parameter algorithm under the well-established hypothesis that W[1] $\neq$ FPT. A parameterized reduction from $\mathcal{P}$ to a parameterized problem $\mathcal{Q}$ is a function which:

- maps **YES**-instances to **YES**-instances and **NO**-instances to **NO**-instances,
- is computable in time $f(p) \cdot n^{\mathcal{O}(1)}$, where $f$ is a computable function, and
- ensures the parameter of the output instance can be upper-bounded by some function of the parameter of the input instance.

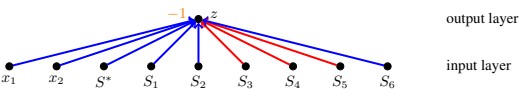

Figure 2: An illustration of the reduction behind Theorem 1 for the universe $U = [6]$ and the set family $\mathcal{F}$ with sets $S_1 = \{1, 4, 5\}, S_2 = \{2, 3\}, S_3 = \{1, 6\}, S_4 = \{2, 5\}, S_5 = \{3, 5\}, S_6 = \{6\}$ with an exact set cover $\mathcal{S} = \{S_1, S_2, S_6\}$. In the solution corresponding to $\mathcal{S}$, each red arc has weight 0 and each blue arc has weight 1. The orange number is the bias of the output neuron.

## 3  LOWER BOUNDS FOR $d$-QNNT

In this section, we show that 2-QNNT remains intractable in highly restrictive settings. First, in Theorem 1, we establish NP-hardness even if the architecture has no hidden neuron, only one output neuron, and for training without error. Note that Theorem 1 implies NP-hardness even when the combined parameter width + depth + $\ell$ + $\omega$ is upper-bounded by a constant. Naturally, the corresponding reduction requires the output neurons to have an arbitrarily large degree. One could hence hope that architectures with constant maximum degree can be trained efficiently. In Theorem 2, we show that this is not possible by establishing NP-hardness for this setting.

In both the reductions that underlie Theorems 1 and 2 the number of input neurons is large and in particular not upper-bounded by a function of the parameters. Hence, one could hope that a small or even constant number of inputs allows for efficient training. We show that this is not the case either. First, in Theorem 3, we provide W[2]-hardness for $\alpha$ even if there is no hidden layer. Second, in Theorem 4, we show that 2-QNNT remains NP-hard even if there are only 2 inputs and 1 hidden layer. Altogether, these results yield the lower bounds depicted in Figure 1.

**Theorem 1.** 2-QNNT *is* NP-*hard even when restricted to instances where $\ell = 0$ and architectures with a single output neuron and no hidden neuron.*

*Proof.* We provide a reduction from the NP-hard EXACT SET COVER problem (Karp, 1972) where the input consists of a universe $U$, and a family $\mathcal{F}$ of subsets over $U$. The goal is to find a subset $\mathcal{S} \subseteq \mathcal{F}$ such that $\mathcal{S}$ is a partition of $U$, that is, 1) $\bigcup_{S \in \mathcal{S}} S = U$ and 2) $S_1 \cap S_2 = \emptyset$ for each $S_1, S_2 \in \mathcal{S}$.

**Construction.** We construct an equivalent instance $I$ of 2-QNNT as follows; see Figure 2 for an illustration.

*Description of the architecture $G$.* Abusing notation, for each set $F \in \mathcal{F}$ we create a *set input neuron $F$*. Moreover, we add 3 more *dummy input neurons $S^*$, $x_1$,* and *$x_2$*, respectively. Finally, we add one output neuron $z$ and add an arc from each input neuron to the unique output neuron $z$.

*Description of the data set.* For each element $u \in U$ we add two *element data points*: $d_u^1$ and $d_u^2$: both have value 1 in each input corresponding to a set containing $u$ and value 0 in dummy inputs $x_1$ and $x_2$. Moreover, $d_u^1$ has value 0 in dummy input $S^*$ and value 0 in output $z$, and $d_u^2$ has value 1 in dummy input $S^*$ and value 1 in output $z$. Finally, we add three further data points: *dummy data points $d_{01}, d_{10}$, and $d_{11}$.* All three have value 0 in each set input and in dummy input $S^*$. Moreover, $d_{01}$ has values $x_1 = 0$, $x_2 = 1$ and output value 0, $d_{10}$ has values $x_1 = 1$, $x_2 = 0$ and output value 0, and $d_{11}$ has values $x_1 = 1$, $x_2 = 1$ and output value 1.

Finally, we set $\ell = 0$. To complete the proof, it remains to establish correctness. $(\star)$

**Correctness.** We verify that $(U, \mathcal{F})$ has an exact set cover $\mathcal{S}$ if and only if $I$ is a yes-instance of 2-QNNT.

$(\Rightarrow)$ Let $\mathcal{S}$ be an exact cover for $(U, \mathcal{F})$. We now argue that assigning the unique output neuron $z$ a bias of $-1$, a weight of 1 to each arc starting from a set $S \in \mathcal{S}$ or any dummy input, and weight 0 to any remaining arc, yields a solution to $I$. The dummy data points clearly yield the desired output. Moreover, for any element $u \in U$ the output of data point $d_u^1$ is 0 since there is exactly one set $S \in \mathcal{S}$ containing $u$. By the same argument data point $d_u^2$ yields output 1 since additionally dummy input $S^*$ has value 1.

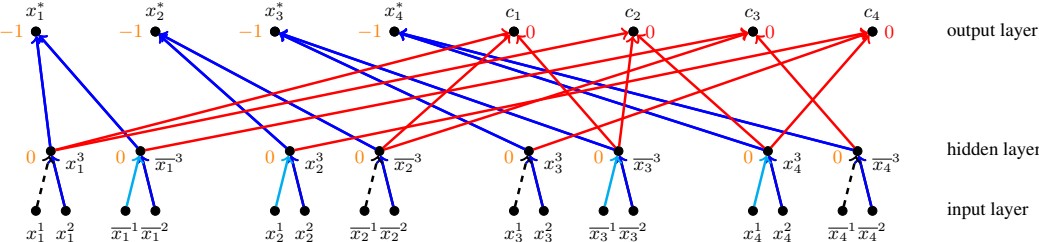

Figure 3: An illustration of the reduction behind Theorem 2 for the formula $\Phi$ with clauses $c_1 = x_1 \vee \overline{x_2} \vee \overline{x_3}$, $c_2 = x_1 \vee \overline{x_3} \vee x_4$, $c_3 = \overline{x_1} \vee \overline{x_2} \vee \overline{x_4}$, and $c_4 = x_2 \vee x_3 \vee x_4$ (here the property that each literal appears exactly twice is dropped) with a satisfying assignment $\mathcal{A}$ with $\{x_2, x_4\} \mapsto \texttt{true}$ and $\{x_1, x_3\} \mapsto \texttt{false}$. In an optimal solution $S$ the blue arcs are the edges of the fake variable gadgets. The all/true/false fake data point imply that all these blue arcs have weight 1 and also that the biases shown in orange. These gadget enforce the selection of an assignment of the variables. Moreover, in $S$ we can assume without loss of generality that the weight of the red arcs is 1 and that the biases shown in red are 0. The cyan arcs correspond to the assignment $\mathcal{A}$ and have weight 1 and the dashed black arcs have weight 0. Moreover, $\mathcal{A}$ needs to be satisfying because of the red part.

($\Leftarrow$) First observe that all dummy data points can only yield the desired outputs if the bias of the unique output $z$ is 1, and the weights of the arcs $(x_1, z)$ and $(x_2, z)$ is 1. Now, we set $\mathcal{S} := \{S \in \mathcal{F} : \texttt{weight}(S, z) = 1\}$ and we claim that $\mathcal{S}$ is an exact set cover for $(U, \mathcal{F})$: 1) Since set data point $d_u^1$ yields output 0, at most one set can contain element $u$. 2) Since set data point $d_u^2$ yields output 1 and the unique input which has value 1 for $d_u^2$ which is not a set input is the dummy input $S^*$, we observe that at least one set has to contain element $u$. This now implies that each element is covered exactly once and thus $\mathcal{S}$ is an exact set cover. $\square$

We note that one could also obtain Theorem 1 by carefully adapting the hardness proof of Schmitt (2004, Theorem 7) to our setting. However, the reduction we provide here is simpler, self-contained, and additionally also implies W[1]-hardness with respect to the number of arcs with weight one in the solution. We continue by stating the hardness for constant-degree architectures; since this result is not central to our complexity landscape (see Figure 1), we defer its proof to the appendix.

**Theorem 2.** *2-QNNT is NP-hard even when restricted to instances where $\ell = 0$, $|\mathcal{D}| \leq 4$, and architectures with only one hidden layer, maximum outdegree 3, and maximum indegree 2.*

*Proof.* We present a reduction from the NP-complete $(3, B2)$-SAT problem (Berman et al., 2003), a variant of 3-SAT where one is given a CNF formula $\Phi$ on variables $x_1, \ldots, x_n$ where each of the $m$ clauses contains exactly three literals and each literal $x_i$ and $\overline{x_i}$ occurs exactly twice in $\Phi$.

**Construction.** We construct an equivalent instance $I$ of $d$-QNNT as follows. For an illustration, see Figure 3.

*Description of architecture $G$.* For each literal $\ell_i$ (note that $\ell_i = x_i$ or $\ell_i = \overline{x_i}$) we create 3 neurons: an *original input neuron* $\ell_i^1$, a *fake input neuron* $\ell_i^2$, and a *hidden neuron* $\ell_i^3$. The inputs are the union of all original and fake input neurons. Moreover, for each variable $x_i$ we create a *variable output neuron* $x_i^*$ and for each clause $c_j$ we create a *clause output neuron* $c_j$. The outputs are the union of all variable and clause output neurons. Note that we have $7n + m$ neurons in total and $4n$ of those are inputs and $n + m$ of them are outputs.

We connect the neurons as follows: We add the arcs $(\ell_i^1, \ell_i^3)$ and $(\ell_i^2, \ell_i^3)$. Let $x_i$ be the variable corresponding to literal $\ell_i$ and let $C$ be the set of literals containing literal $\ell_i$. We add the arcs $(\ell_i^3, x_i^*)$, and $(\ell_i^3, c)$ for any $c \in C$. This completes the construction of the architecture $G$. Note that any neuron in $G$ has an indegree of at most 3, matched by any clause output and out-degree at most 3, matched by any hidden neuron since by our assumption each literal occurs exactly twice in $\Phi$, respectively.

*Description of data set.* In the following, we use the notation $(a_1, a_2, a_3, a_4) \mapsto (a_5, a_6)$ for the data points. Entries $a_1$ to $a_4$ correspond to the inputs and entries $a_5$ and $a_6$ correspond to the outputs.

More precisely, **(1)** $a_1$ corresponds to all original inputs corresponding to a positive literal, **(2)** $a_2$ corresponds to all original inputs corresponding to a negative literal, **(3)** $a_3$ corresponds to all fake inputs corresponding to a positive literal, **(4)** $a_4$ corresponds to all fake inputs corresponding to a negative literal, **(5)** $a_5$ corresponds to all variable outputs, and **(6)** $a_6$ corresponds to all clause outputs. Whenever we put a 0 or 1 in any of these entries, we mean that all corresponding inputs/outputs receive value 0 or 1, respectively.

We add 4 data points: **(1)** The *all fake data point* with $(0, 0, 1, 1) \mapsto (1, 1)$. **(2)** The *true fake data point* with $(0, 0, 1, 0) \mapsto (0, c_{\texttt{true}})$, where an output entry $c_j$ of $c_{\texttt{true}}$ is 1 if and only if clause $c_j$ contains at least one positive literal, and 0 otherwise. **(3)** The *false fake data point* with $(0, 0, 0, 1) \mapsto (0, c_{\texttt{false}})$, where an output entry $c_j$ of $c_{\texttt{false}}$ is 1 if and only if clause $c_j$ contains at least one negative literal, and 0 otherwise. **(4)** The *assignment data point* with $(1, 1, 0, 0) \mapsto (0, 1)$.

Finally, we set $\ell = 0$. This finishes the description of our $d$-QNNT instance $I$.

**Intuition.** The arcs from the original inputs to the hidden neurons model a variable assignment, that is, at most one of the arcs $(x_i^1, x_i^3)$ and $(\overline{x_i}^1, \overline{x_i}^3)$ can have weight 1. This is enforced with the fake inputs, the hidden neurons, and the variable outputs together with the all/true/fake data points. More precisely, these neurons imply that all blue arcs of Figure 3 have weight 1, that the hidden neurons have bias 0, and that the variable output neurons have bias $-1$. Moreover, it is safe to assume that any red arc of Figure 3 has weight 1 and that the bias of any clause output neuron is 0, as we show. This then implies that the variable assignment needs to satisfy formula $\Phi$.

**Correctness.** We now verify that $\Phi$ is satisfiable if and only if $I$ is a yes-instance of $d$-QNNT.

$(\Rightarrow)$ Let $\mathcal{A} : (x_i)_{i \in [n]} \to \{\texttt{true}, \texttt{false}\}$ an assignment to the variables which satisfies $\Phi$. We now show how to set the functions `weight` and `bias` such that there is no error, also see Figure 3. **(1)** We start with the `weight` function: The arcs incident to any fake input neuron, as well as the arcs incident to any output neuron have weight 1. It remains to consider the arcs incident to original input neurons. If $\mathcal{A}(x_i) = \texttt{true}$, then the arc $(x_i^1, x_i^3)$ gets weight 1 and the arc $(\overline{x_i}^1, \overline{x_i}^3)$ gets weight 0, and otherwise if $\mathcal{A}(x_i) = \texttt{false}$, then the arc $(x_i^1, x_i^3)$ gets weight 0 and the arc $(\overline{x_i}^1, \overline{x_i}^3)$ gets weight 1. **(2)** We continue with the `bias` function: The bias of any hidden neuron and any clause output neuron is 0, and the bias of any variable output neuron is $-1$.

It remains to verify that there is no error. We consider each data point individually:

1. Consider the all fake data point. Since all arcs incident to any fake input have weight 1 and since any hidden neuron has bias 0, we observe that any hidden neuron is active. Consequently, also all output neurons are active, which is correct.
2. Consider the true fake data point. Similarly to the all fake data point, we observe that all hidden neurons corresponding to positive literals are active but all hidden neurons corresponding to negative literals are inactive. Consequently, each variable output is 0. Moreover, a clause output neuron $c_j$ is active if and only if clause $c_j$ contains a positive literal which matched the definition of vector $c_{\texttt{true}}$. Thus, the true fake data point is evaluated correctly.
3. The argumentation for the false fake data point is analog to the true fake data point by swapping the roles of positive and negative literals.
4. Consider the assignment data point. If $\mathcal{A}(x_i) = \texttt{true}$, then hidden neuron $x_i^3$ is active and hidden neuron $\overline{x_i}^3$ is inactive, and otherwise if $\mathcal{A}(x_i) = \texttt{false}$, then hidden neuron $x_i^3$ is inactive and hidden neuron $\overline{x_i}^3$ is active. Consequently, all variable output neurons are inactive. Moreover, since $\mathcal{A}$ is satisfying, all clause output neurons are active and thus the assignment data point is evaluated correctly.

Hence, there is no error.

$(\Leftarrow)$ Let `weight` and `bias` be functions such that the resulting neural network $\bar{G}$ has no errors. We now argue how to construct a satisfying assignment for $\Phi$. By the *fake variable gadget of $x_i$* we mean the induced subnetwork of the 5 neurons corresponding to variable $x_i$ and the two associated fake literals $x_i$ and $\overline{x_i}$, that is, the neurons $x_i^z, \overline{x_i}^z$ for $z \in \{2, 3\}$, and $x_i^*$; see also Figure 3.

We proceed as follows: In the first step, we argue that all arc weights in any fake variable gadget has to be 1, that the bias of any hidden neuron is 0, and that the bias of any variable output neuron is $-1$. Second, we show that we can safely assume that all arcs from any hidden neuron to any clause output

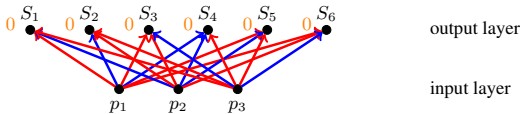

Figure 4: An illustration of the reduction behind Theorem 3 for the universe $U = [6]$ and the set family $\mathcal{F}$ with sets $S_1 = \{1, 4, 5\}, S_2 = \{2, 3\}, S_3 = \{1, 6\}, S_4 = \{2, 5\}, S_5 = \{3, 5\}, S_6 = \{6\}$ and $k = 3$ and with a hitting set $S = \{2, 5, 6\}$. In the solution corresponding to $S$, inputs $p_1$, $p_2$ and $p_3$ are associated with elements 2, 5 and 6, respectively. Moreover, each red arc has weight 0 and each blue arc has weight 1. The orange numbers are the biases of the output neurons.

neuron have weight 1, and that all clause output neurons have bias 0. In the final step, we argue that the weights of the arcs incident to the original inputs correspond to a satisfying assignment of $\Phi$.

*Step 1.* Consider the all fake input point $q$. Recall that in $q$ only all fake inputs have value 1 and that all variable outputs have value 1. Now, consider an arbitrary but fixed variable $x_i$ and its associated fake variable gadget. Note that there are exactly two paths from active input neurons to the active variable output neuron $x_i^*$: $p_1 := (x_i^2, x_i^3, x_i^*)$ and $p_2 := (\overline{x_i}^2, \overline{x_i}^3, x_i^*)$. Since in $q$ the variable output neuron $x_i^*$ is active, one at least one of the paths $p_1$ or $p_2$ all arc weights are 1 and the bias of the hidden neuron is 0. Without loss of generality, we assume that this is the case for $p_1$. Next, observe that in the true fake data point the fake input $x_i^2$ is also active, but the variable output neuron $x_i^*$ is inactive. Consequently, neuron $x_i^*$ has a bias of $-1$. Now, again consider the all fake data point $q$: In order to activate neuron $x_i^*$ also the weights of all arcs in $p_2$ have to be 1 and also the bias of neuron $\overline{x_i}^3$ needs to be 0. Thus, Step 1 is accomplished.

*Step 2.* We now argue that we can safely change the weight of any arc incident to a clause output neuron $c_j$ from 0 to 1 and that we can also safely change the bias of any clause output neuron $c_j$ from $-1$ to 0: Note that such a change can only be unfavorable for a data point where $c_j$ has value 0. Consequently, this can only affect the true (and the false) fake data point. More precisely, only clause output neurons corresponding to clauses which do not contain any true (false) literal have value 0 in the true (false) fake data point. Hence, the two changes to not change the value of 0 of any such clause output neuron and thus Step 2 is accomplished.

*Step 3.* Consider the arcs $(x_i^1, x_i^3)$ and $(\overline{x_i}^1, \overline{x_i}^3)$ incident to the original inputs, and the assignment data point $q$. Note that at most one of these arcs can have weight 1: If both have weight 1 then for data point $q$ both hidden neurons $x_i^3$ and $\overline{x_i}^3$ are active, and thus also the variable output neuron $x_i^*$, contradicting the correct value of 0 for that output neuron. We now define a variable assignment $\mathcal{A}$: $\mathcal{A}(x_i) := \texttt{true}$ if $\texttt{weight}(x_i^1, x_i^3) = 1$, and $\mathcal{A}(x_i) := \texttt{false}$ otherwise.

Observe that $\mathcal{A}$ is satisfying $\Phi$: Consider an arbitrary but fixed clause with literals $\ell_1, \ell_2$, and $\ell_3$. Note that $p_z := (\ell_z^1, \ell_z^3, c_j)$ is the path from the original input neuron $\ell_z^1$ to the clause output neuron $c_j$ for any $z \in [3]$ and that there is no other path from any input neuron to output neuron $c_j$. Since in the assignment data point $q$ the clause output neuron $c_j$ has value one, the weight of all arcs on one path $p_z$ has to be 1. Without loss of generality, we assume that is the case for $p_1$. Consequently, by our definition of $\mathcal{A}$, literal $\ell_1$ satisfies $c_j$ and hence the statement is proven. $\square$

Next, we establish W-hardness w.r.t. the number $\alpha$ of inputs even if there is no hidden layer.

**Theorem 3.** *Even if the network has no hidden neuron, 2-QNNT is W[2]-hard when parameterized by the number $\alpha$ of input nodes, even when restricted to architectures with no hidden neurons.*

*Proof.* We present a reduction from the HITTING SET (HS) problem where the input consists of a universe $U$, a family $\mathcal{F}$ of subsets over $U$, and an integer $k$. The goal is to find a subset $S \subseteq U$ (called a *hitting set*) of size $k$ such that $S$ contains at least one element of each set in the family, that is, $S \cap F \neq \emptyset$ for any $F \in \mathcal{F}$. HS is W[2]-hard parameterized by $k$ (Cygan et al., 2015).

**Construction.** We construct an instance $I$ of 2-QNNT as follows. For an illustration, see Figure 4. *Description of the architecture $G$.* We create $k$ input neurons $p_1, \ldots, p_k$. Abusing notation, for each set $F \in \mathcal{F}$ we create one *set output neuron $F$*. We add arcs between every input and output neuron. *Description of the data set.* For each element $u \in U$ we add $k$ *element $u$ data points $d_u^1, \ldots, d_u^k$*.

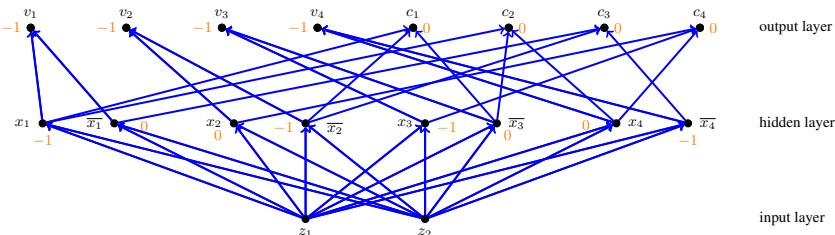

Figure 5: An illustration of the reduction behind Theorem 4 for the formula $\Phi$ with clauses $c_1 = x_1 \vee \overline{x_2} \vee \overline{x_3}$, $c_2 = x_1 \vee \overline{x_3} \vee x_4$, $c_3 = \overline{x_1} \vee \overline{x_2} \vee \overline{x_4}$, and $c_4 = x_2 \vee x_3 \vee x_4$ with a satisfying assignment $\mathcal{A}$ with $\{x_2, x_4\} \mapsto$ `true` and $\{x_1, x_3\} \mapsto$ `false`. In an optimal solution all arcs have weight 1. The biases of of a solution corresponding to $\mathcal{A}$ are shown in orange.

Element $u$ data point $d_u^i$ has value 1 in input $p_i$ and value 0 in each other input. Moreover, $d_u^i$ has value 1 in each set output $F$ such that $u \in F$. Thus, $d_u^i$ has value 0 in each set output $F'$ such that $u \notin F'$. Observe that the $k$ element $u$ data points all have the same output but they have pairwise different inputs. Then, we add a *verifier data point* $d^*$ which has value 1 in each input and in each output. In the following, we say that two data points $d_1$ and $d_2$ have the same *type* if the input values of $d_1$ and $d_2$ are pairwise identical. Note that we have exactly $k + 1$ distinct types of data points.

Finally, we set $\ell := k \cdot (|U| - 1)$. To complete the proof, it remains to establish correctness. $(\star)$

**Correctness.** We verify that $(U, \mathcal{F})$ has a hitting set $S$ of size $k$ if and only if $I$ is a yes-instance of 2-QNNT. Before we prove the correctness, we make the following crucial observation about $I$: Since at most one data point of any type of data points can be correctly computed, in total at most $k + 1$ data points can be correctly computed. Since we have $k \cdot |U| + 1$ data points, and $\ell = k \cdot (|U| - 1)$, exactly one data point of each type has to be classified correctly.

The intuition is that $k$ element data points need to be computed correctly. These then correspond to a set $S$ of elements. Since also the verifier data point needs to be correctly computed, this then implies that $S$ has to be a hitting set.

We let $\mathcal{F}(u_i) := \{F \in \mathcal{F} : u_i \in F\}$ denote the family of subsets of $\mathcal{F}$ which contain element $u_i \in U$.

$(\Rightarrow)$ Let $S$ be a hitting set of size at most $k$ for $(U, \mathcal{F})$. Let $u_1, \ldots, u_k$ be the elements of $S$ in some arbitrary but fixed order. For any $u_i \in S$ we set $\texttt{weight}(u_i, x_i) = 1$ for any $x_i \in \mathcal{F}(u_i)$. For any other arc $e$, we set $\texttt{weight}(e) = 0$. Observe that this yields a correct computation of element $u_i$ data point $d_{u_i}^i$ for any $u_i \in S$. Moreover, since $S$ is a hitting set, also the verifier data points gets computed correctly. Consequently, $k + 1$ data points are computed correctly, and using the observation we conclude that $I$ is a yes-instance.

$(\Leftarrow)$ According to the observation, exactly one data point of each type has to be computed correctly. Thus, the verifier data point $d^*$ has to be computed correctly, and for any $i \in [k]$ exactly one data point which has value 1 in input $p_i$ and value 0 in each other input. Since each data point having these inputs, is an element $u'$ data point $d_{u'}^i$ for some $u' \in U$, we conclude that there exists some element $u \in U$ such that $d_u^i$ gets correctly computed. By $u_i$ we denote the element corresponding to the correctly computed data point $d_u^i$. Consequently, we have $\texttt{weight}(u_i, F_i) = 1$ for each $F_i \in \mathcal{F}(u_i)$ and $\texttt{weight}(u_i, F_i) = 0$ for each $F_i' \notin \mathcal{F}(u_i)$. Due to the correct computation of the verifier data point $d^*$, we observe that $\texttt{weight}(u, F) = 1$ for each $F \in \mathcal{F}$ and thus the set $S := \{u_i \in U : d_u^i \text{ is correctly computed}\}$ is a hitting set of size $k$ for $(U, \mathcal{F})$. $\square$

For our fourth lower bound, we use a "compressed" version of the construction behind Theorem 2 to obtain NP-hardness for only 2 input nodes and 3 data points.

**Theorem 4.** *2-QNNT is NP-hard even if $\alpha = 2$, $\ell = 0$, $|\mathcal{D}| = 3$, and depth $= 1$.*

*Proof.* We present a reduction from 3-SAT (Karp, 1972), where one is given a CNF formula $\Phi$ on variables $x_1, \ldots, x_n$ and a set of $m$ clauses each consisting of precisely three literals.

**Construction.** We construct an equivalent instance $I$ of 2-QNNT as follows; see Figure 5 for an illustration.

*Description of architecture $G$.* We create two input neurons $z_1$ and $z_2$. For each of the two literals of a variable $x_i$ with $i \in [n]$, we create two *hidden neurons $x_i$ and $\overline{x_i}$ associated with variable $x_i$*. Thus, we create $2n$ hidden neurons. Moreover, we create a *variable output neuron $v_i$ associated with variable $x_i$* for each variable $x_i$. Also, we add one *clause output neuron $c_j$* for each clause of $\Phi$. Thus, we create $n + m$ output neurons.

We add an arc from each input neuron to each hidden neuron. Next, we add an arc from each of the two hidden neurons $x_i$ and $\overline{x_i}$ associated with variable $x_i$ to the variable output neuron $v_i$ associated with variable $x_i$. Finally, for each clause $c_j$ consisting of literals $p_1, p_2$, and $p_3$, we add the arcs $(p_h, c_j)$ for each $h \in [3]$.

*Description of data set.* Here, we use the notation $(z_1, z_2) \mapsto (V, C)$ for the data points, where $z_1$ and $z_2$ are numbers referring to the inputs, and $V$ and $C$ are vectors referring to the outputs. More precisely, $V$ has length $n$, and the $i$-th entry corresponds to the variable output neuron $v_i$, and $C$ has length $m$, and the $j$-th entry corresponds to the clause output neuron $c_j$. Whenever we put a 0 or a 1 in any of the three vectors, we mean that all corresponding outputs receive value 0 or 1, respectively.

We add 3 data points: **(1)** The *verifier 1 data point* with $(1, 0) \mapsto (0, 1)$, **(2)** the *verifier 2 data point* with $(0, 1) \mapsto (0, 1)$, and **(3)** the *choice data point* with $(1, 1) \mapsto (1, 1)$.

Finally, we set $\ell := 0$.

Recall that we say that given a data point $p$ a neuron $q$ is *active* if in the evaluation of $p$, the neuron $q$ receives a positive activation; otherwise, it is *inactive*.

**Intuition.** The idea is that when considering the verifier 1 data point, the active hidden neurons correspond to a satisfying variable assignment. We achieve this with the variable output neurons: If both hidden neurons $x_i$ and $\overline{x_i}$ associated with a variable $x_i$ are active for the verifier 1 data point, then since the value of the variable output neuron $v_i$ associated with $x_i$ needs to be 0 and since $x_i$ and $\overline{x_i}$ are the unique neighbors of $v_i$ this then implies that the value of $v_i$ for the choice data point is also 0, and not 1 as desired, yielding an error.

**Correctness.** We now verify that $\Phi$ is satisfiable if and only if $I$ is a yes-instance of 2-QNNT.

($\Rightarrow$) Let $\mathcal{A} : (x_i)_{i \in [n]} \to \{\texttt{true}, \texttt{false}\}$ be an assignment to the variables which satisfies $\Phi$. We now show how to set the functions $\texttt{weight}$ and $\texttt{bias}$ such that there is no error, also see Figure 5. **(1)** First, we set the $\texttt{weight}$ of any arc in $G$ to 1. **(2)** Second, we set the biases: **(a)** For each clause output neuron $c_j$, we set $\texttt{bias}(c_j) = 0$. **(b)** For each variable output neuron $v_i^1$, we set $\texttt{bias}(v_i^1) = 0$. **(c)** For each clause output neuron $v_i^2$, we set $\texttt{bias}(v_i^2) = -1$. **(d)** Finally, for the two hidden neurons $x_i$ and $\overline{x_i}$ associated with variable $x_i$, we set $\texttt{bias}(x_i) = 0$ and $\texttt{bias}(\overline{x_i}) = -1$ if $\mathcal{A}(x_i) = \texttt{true}$, and otherwise we set $\texttt{bias}(x_i) = -1$ and $\texttt{bias}(\overline{x_i}) = 0$ if $\mathcal{A}(x_i) = \texttt{false}$. By $\bar{G}$ we denote the resulting network.

It remains to verify that all 3 data points get computed correctly.

First, we consider the verifier 1 and verifier 2 data points: Observe that the hidden neuron $x_i$ is active and the hidden neuron $\overline{x_i}$ is inactive if $\mathcal{A}(x_i) = \texttt{true}$ and, the hidden neuron $x_i$ is inactive and the hidden neuron $\overline{x_i}$ is active otherwise if $\mathcal{A}(x_i) = \texttt{false}$, respectively. Consequently, each variable output $v_i$ yields output 0. Moreover, since $\mathcal{A}$ satisfied $\Phi$, we conclude that also each clause output yields output 1. Consequently, the verifier 1 and verifier 2 data points are computed correctly.

Second, we consider the choice data point: Observe that all hidden neurons are active and consequently also all output neurons are active showing that also the choice data point in correctly computed. Thus, $\bar{G}$ has no errors.

($\Leftarrow$) Let $\texttt{weight}$ and $\texttt{bias}$ be functions such that the resulting network $\bar{G}$ has no errors. We now argue how to construct a satisfying assignment $\mathcal{A}$ for $\Phi$. Since there is no error, the verifier 1 data point needs to be computed correctly. Observe that for any variable $x_i$ *at most one* of the two hidden neurons $x_i$ and $\overline{x_i}$ associated with variable $x_i$ is active for the verifier 1 data point: Assume towards a contradiction that this is not that case, that is, that there exists a variable $x_i$ such that *both* hidden neurons $x_i$ and $\overline{x_i}$ associated with variable $x_i$ are both active for the verifier 1 data point. Again, since the verifier 1 data point is correctly computed, the variable output neuron $v_i$ has value 0. Recall that $v_i$ is incident with the arcs $(x_i, v_i)$ and $(\overline{x_i}, v_i)$ and that both are active for the verifier 1 data

point. Thus, for the choice data point the variable output neuron $v_i$ will also yield value 0, yielding a contradiction to the fact that there is no error since the output of $v_i$ should be 1 for the choice data point.

Let $X \subseteq [n]$ be the set of indices such that *exactly one* hidden neuron $x_i$ or $\overline{x_i}$ associated with variable $x_i$ is active for the verifier 1 data point. We now define a partial assignment $\mathcal{A}$ for the variables with indices in $X$ as follows: We set $\mathcal{A}(x_i) = \texttt{true}$ is and only if $x_i$ is active, and we set $\mathcal{A}(x_i) = \texttt{false}$ is and only if $\overline{x_i}$ is active. To see that $\mathcal{A}$ satisfies $\Phi$, note that for the verifier 1 data point each clause output needs to have value 1. Also, recall that clause $c_j$ is incident with the arcs $(p_h, c_j)$ where $p_h$ for $h \in [3]$ are the 3 literals of $c_j$. Since there is no error, at least one of the hidden neurons $p_h$ needs to be active for the verifier 1 data point and $\texttt{weight}(p_h, c_j) = 1$ for at least one $h \in [3]$, showing that clause $c_j$ is satisfied by literal $p_h$. Note that $\mathcal{A}$ can be extended to an assignment $\mathcal{A}'$ of all variables by simply assigning $\texttt{true}$ to any remaining variable. Since already $\mathcal{A}$ was satisfying $\Phi$, assignment $\mathcal{A}'$ satisfies $\Phi$ as well. $\qquad\square$

## 4 FIXED-PARAMETER TRACTABILITY

In this section we prove our tractability results for parameter combinations that include the width, treewidth, and number $\alpha$ of input neurons. We begin by showing a structural result (Lemma 1) that states that there is always a solution that has upper-bounded degree in the sense that, for each neuron, there is only a bounded number of incoming arcs with nonzero weights. We then use Lemma 1 to prove tractability of $d$-QUANTIZED RELU-ACTIVATED NEURAL NETWORK TRAINING ($d$-QNNT) without error with respect to the treewidth and number $\alpha$ of input neurons (Lemma 3). Then we show how to lift this result to training with nonzero error bounds and how the treewidth results imply the corresponding results for the width.

Consider a neuron $v$ in a neural network. Define the *non-zero in-neighbors* of $v$ to be the in-neighbors $u$ of $v$ such that $\texttt{weight}(uv) \neq 0$. The *non-zero indegree* of $v$ is the number of non-zero in-neighbors.

**Lemma 1.** *Let $G$ be an architecture and $\mathcal{D}$ a data set with $p$ distinct input vectors. If there is a neural network over $G$ with zero error on $\mathcal{D}$, then there is a neural network $\bar{G}$ over $G$ with zero error on $\mathcal{D}$ such that for each neuron $v$ in $\bar{G}$ the number of non-zero in-neighbors of $v$ is at most $(dp)^{\mathcal{O}(p)}$.*

We prove Lemma 1 by using Steinitz' Lemma, stated as follows.

**Lemma 2** (Steinitz' Lemma (Steinitz, 1913; Sevast'janov, 1994))**.** *Let $\| \cdot \|$ be an arbitrary norm on $\mathbb{R}^d$. Let $x_1, \ldots, x_m \in \mathbb{R}^d$ such that $\sum_{i \in [m]} x_i = 0$ and for each $i \in [m]$ we have $\|x_i\| \leq 1$. Then there exists a permutation $\pi \in S_m$ such that all prefix sums have norm at most $d$. That is, for each $k \in [m]$ we have $\| \sum_{j \in [k]} x_{\pi(j)} \| \leq d$.*

The idea of the proof of Lemma 1 is as follows. Consider a neuron $v$ in a solution network. We can collect the activations of $v$ for each input vector in a vector $\vec{s} \in (\mathbb{Z}_d)^p$. Assume for simplicity that we don't have ReLU activations and instead simply pass through the weighted sum of the activations of the in-neighbors and, furthermore, each of the summed activations is in $(\mathbb{Z}_d)^p$. Then, $\vec{s}$ is a small-norm vector and it is obtained as a sum of small-norm vectors. Steinitz' Lemma tells us that we can reorder the vectors such that each prefix sum has small norm. This means that, if there are many non-zero in-neighbors to $v$, then at least one prefix sum occurs twice. This means that the vectors in between these two identical sums sum to zero and we can simply set their corresponding arc weights to zero without changing the activation of $v$. Care must be taken to preserve the ReLU activations and boundaries of $(\mathbb{Z}_d)^p$ and to ensure that all vectors in the sum have small norm.

*Proof of Lemma 1.* Assume that there is a neural network $\bar{G}'$ over $G$ with zero error on $\mathcal{D}$. Consider an arbitrary neuron $v$ with more than $2 \cdot (2d^2 p + 1)^p + 1$ non-zero in-neighbors. Let $q$ be the number of such in-neighbors of $v$ and label them $u_1, \ldots, u_q$. We show that we can set the weight of at least one arc from a non-zero in-neighbor to 0 without changing the activation value of $v$ for each input vector.

For each non-zero in-neighbor $u_i$, $i \in [q]$, let $\vec{y}^{(i)} \in (\mathbb{Z}_d)^p$ be a vector such that for each $j \in [p]$ the $j$th entry $y_j^{(i)}$ of $\vec{y}^{(i)}$ is the activation value of $u_i$ on input of the $j$th input vector multiplied

with `weight`$(u_i v)$. Similarly, let $\vec{s} \in \mathbb{Z}^p$ be the vector containing the pre-activation values that $v$ receives from all in-neighbors for each input vector.

We have $\sum_{i=1}^{q} \vec{y}^{(i)} = \vec{s}$. Note that $\vec{s}$ may contain arbitrarily large values. To obtain a sum of small-norm vectors we replace $\vec{s}$ by a sequence of unit vectors and $\vec{t}$ which we define now. Intuitively, for each large or small entry $\vec{s}_j$, entry $\vec{t}_j$ will contain a lower or upper bound for the pre-activation value of $v$ received from in-neighbors such that the value of $v$ remains the same, even if the pre-activation value is reduced or increased to the corresponding bound. Precisely, for each $j \in [p]$, if $\vec{s}_j$ is negative we put entry $\vec{t}_j := \max\{-d, \vec{s}_j\}$ and otherwise we put entry $\vec{t}_j := \min\{d, \vec{s}_j\}$. (We could replace $d$ in the maximum and minimum by a floor or ceiling of $\frac{d-1}{2}$ but the change in the bound is immaterial and the expressions are simpler.) Note that, indeed, if the pre-activation value of $v$ received from in-neighbors for each input vector is as defined in $\vec{t}$, then the value of $v$ will be the same as if the pre-activation value of $v$ received from in-neighbors would be $\vec{s}$ because the bias of $v$ is between $-\lfloor \frac{d-1}{2} \rfloor$ and $\lceil \frac{d-1}{2} \rceil$.

We now replace $\vec{s}$ by unit vectors and $\vec{t}$. For each $j \in [p]$ such that $\vec{s}_j < \vec{t}_j$ (in particular, this means $\vec{s}_j < 0$) define $|\vec{s}_j - \vec{t}_j|$ *dummy vectors* whose $j$th entry is $-1$ and all other entries are $0$. Analogously, for each $j \in [p]$ such that $\vec{s}_j > \vec{t}_j$ (in particular, $0 > \vec{s}_j$) define $\vec{s}_j - \vec{t}_j$ *dummy vectors* whose $j$th entry is $1$ and all other entries are $0$. We say that these dummy vectors *correspond to* the $j$th input vector. Let $\vec{e}^{(1)}, \ldots, \vec{e}^{(r)}$ be the so-defined dummy vectors. We have

$$\sum_{i=1}^{q} (\vec{y}^{(i)}) - \sum_{\ell=1}^{r} (\vec{e}^{(\ell)}) - \vec{t} = 0. \tag{1}$$

We now apply Steinitz' Lemma (Lemma 2). As the norm $\|\cdot\|$ we pick the infinity norm divided by $d^2$ (note that this results in a norm). Thus, since entries of all vectors in Eq. (1) are in absolute at most $d^2$, all these vectors have norm at most 1. By Steinitz' Lemma there is thus a permutation $\pi$ of the vectors in Eq. (1) such that each prefix sum has norm at most $p$. That is, each entry in a vector corresponding to a prefix sum is an integer between $-d^2 \cdot p$ and $d^2 \cdot p$ (as before, these bounds could be tightened at the cost of readability).

Let $\vec{z}^{(1)}, \vec{z}^{(2)}, \ldots$, be the sequence of vectors in Eq. (1) reordered according to $\pi$. Recall that each prefix sum is a $p$-dimensional vector with one of $2d^2p + 1$ entries in each dimension. Since the indegree of $v$ is at least $2 \cdot (2d^2p + 1)^p + 1$, there are at least that many vectors in the sum in total, giving that many prefix sums as well. Thus there are three prefix sums that are exactly the same. Let $h_1, h_2, h_3$ be the corresponding indices, that is, $\sum_{\ell=1}^{h_1} \vec{z}^{(\ell)} = \sum_{\ell=1}^{h_2} \vec{z}^{(\ell)} = \sum_{\ell=1}^{h_3} \vec{z}^{(\ell)}$. Observe that we have $\sum_{\ell=h_1+1}^{h_2} \vec{z}^{(\ell)} = \sum_{\ell=h_2+1}^{h_3} \vec{z}^{(\ell)} = 0$. We now aim to set to zero the weights of the arcs to $v$ from the in-neighbors that correspond to one of these two intervals.

Since the above are two disjoint sequences of vectors, there is one sequence, say the first one, such that $-\vec{t}$ is not contained in it. Thus, all vectors in $\vec{z}^{(h_1+1)}, \ldots, \vec{z}^{(h_2)}$ are either dummy vectors or weighted activation values of in-neighbors of $v$. Let $Q \subseteq [q]$ be the index set of those $\vec{y}^{(i)}$ that are not in $\vec{z}^{(h_1+1)}, \ldots, \vec{z}^{(h_2)}$ and let $R$ be the index set of those $\vec{e}^{(\ell)}$ not in $\vec{z}^{(h_1+1)}, \ldots, \vec{z}^{(h_2)}$. Thus,

$$\sum_{i \in Q} \vec{y}^{(i)} = \vec{t} + \sum_{\ell \in R} \vec{e}^{(\ell)}. \tag{2}$$

Now modify the neural network $\bar{G}'$ by setting to zero all weights of arcs $u_i v$ where $u_i$ is an in-neighbor of $v$ with $i \notin Q$. In this way, we obtain a neural network $\bar{G}$. The pre-activation vector $\vec{s}_{\bar{G}}$ of $v$ in $\bar{G}$ (that is, the vector containing the pre-activation values that $v$ receives from in-neighbors for each input vector) satisfies $\sum_{i \in Q} \vec{y}^{(i)} = \vec{s}_{\bar{G}}$ and thus by Eq. (2) $\vec{s}_{\bar{G}} = \vec{t} + \sum_{\ell \in R} \vec{e}^{(\ell)}$.

The dummy vectors contain $-1$ in dimensions $j$ where $\vec{s}_j < -d = \vec{t}_j$ and $1$ where $\vec{s}_j > d = \vec{t}_j$. Hence, in dimensions $j$ where $\vec{s}_j < -d$ we have $(\vec{s}_{\bar{G}})_j \leq \vec{t}_j$, where $\vec{s}_j > d$ we have $(\vec{s}_{\bar{G}})_j \geq \vec{t}_j$, and otherwise there are no dummy vectors corresponding to $j$ and thus we have $(\vec{s}_{\bar{G}})_j = \vec{t}_j$. Thus, the activation for each input vector of $v$ is the same in $\bar{G}$ and in $\bar{G}'$.

By repeating the argument for each neuron with large number of non-zero in-neighbors we obtain a neural network in which each neuron has less than $2 \cdot (2d^2p + 1)^p + 1$ non-zero in-neighbors, as required. $\

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

$ then we can decide whether there is a neural network over $G$ that learned all input-output pairs correctly by checking whether $D[r, \emptyset, \emptyset, \emptyset, \emptyset] = 1$ (recall that $\chi(r) = \emptyset$). We now sketch how to correctly compute $D$ for each node of $T$ in a bottom-up fashion; the full details are straightforward and partly omitted.

*Leaf node.* If $t$ is a leaf node, let $\chi(t) = \{v\}$. Then $D[t, \texttt{bias}, \texttt{weight}, \texttt{seen}, \texttt{future}] = 1$ if and only if for each $x \in \mathcal{X}$ we have $\texttt{seen}(x, v) = 0$ ($\texttt{weight}$ is empty). (We do not need to verify the correct input and zero-bias of input neurons and the output of output neurons before they are forgotten by the definition of $D$.)

*Introduce node.* Let $t$ be an introduce node with child $t'$ and $\chi(t) = \chi(t') \cup \{v\}$. We put $D[t, \texttt{bias}, \texttt{weight}, \texttt{seen}, \texttt{future}] = 1$ if and only if there exists a state $(\texttt{bias}', \texttt{weight}', \texttt{seen}', \texttt{future}')$ with $D[t', \texttt{bias}', \texttt{weight}', \texttt{seen}', \texttt{future}'] = 1$ such that the following conditions hold.

- *Consistency on old vertices and arcs.* For all $u \in \chi(t')$ and all $x \in \mathcal{X}$ we have $\texttt{bias}(u) = \texttt{bias}'(u)$, $\texttt{seen}(x, u) = \texttt{seen}'(x, u)$, and $\texttt{future}(x, u) = \texttt{future}'(x, u)$; and for all arcs $(a, b) \in E(G)$ with $a, b \in \chi(t')$ we have $\texttt{weight}(a, b) = \texttt{weight}'(a, b)$.

- *Correct seen activation of new neuron.* For all $x \in \mathcal{X}$ we have

$$\texttt{seen}(x, v) = \sum_{u \in (\chi(t') \cap N^-(v))} \texttt{weight}(u, v) \cdot a_u(x),$$

where $N^-(v)$ are the in-neighbors of $v$ and $a_u(x)$ denotes the activation value of neuron $u$. Note that, since $G$ is a DAG, the values $a_u(x)$ for $u \in \chi(t)$ can be computed in topological order from $\texttt{bias}$ and the totals $\texttt{seen}$ and $\texttt{future}$. Observe that, since $\texttt{seen}(x, u) \in \mathbb{Z}_{d^2\Delta}$ the sum is thus capped between $-d^2\Delta$ and $d^2\Delta$. This is correct, since, if there is a solution with non-zero indegree bounded by $\Delta$, restricting this solution to $V_t$ will give a sum that is also within these bounds.

Again, verification of input, output, and bias of input and output neurons is only required when we forget them.

*Forget node.* Let $t$ be a forget node with child $t'$ and $\chi(t) = \chi(t') \setminus \{v\}$. We put $D[t, \texttt{bias}, \texttt{weight}, \texttt{seen}, \texttt{future}] = 1$ if and only if there exists a state $(\texttt{bias}', \texttt{seen}', \texttt{future}', \texttt{weight}')$ with $D[t', \texttt{bias}', \texttt{weight}', \texttt{seen}', \texttt{future}'] = 1$ such that:

- *Projection.* For all $u \in \chi(t)$ and all $x \in \mathcal{X}$,

$$\texttt{bias}(u) = \texttt{bias}'(u), \quad \texttt{seen}(x, u) = \texttt{seen}'(x, u), \quad \texttt{future}(x, u) = \texttt{future}'(x, u).$$

Moreover, for every arc $(a, b) \in E(G)$ with $a, b \in \chi(t)$ we have $\texttt{weight}(a, b) = \texttt{weight}'(a, b)$; all entries of $\texttt{weight}'$ incident to $v$ are dropped.

- *Ensuring all input seen.* For each $x \in \mathcal{X}$ the value $\texttt{future}'(x, v) = 0$.

- *Ensuring correct inputs.* If $v$ is an input neuron, then $\texttt{bias}'(v) = 0$ and for each $x \in \mathcal{X}$ we have $\texttt{seen}'(x, v)$ equal to the activation value specified in $x$.

- *Ensuring correct outputs.* If $v$ is an output neuron, then with total pre-activation $\texttt{seen}'(x, v)$ and bias $\texttt{bias}'(v)$, the activation of $v$ equals the required value, i.e., for

all $x \in \mathcal{X}$ the activation of $v$ coincides with the value specified in the output vector corresponding to $x$.

*Join node.* Let $t$ be a join node with children $t_1, t_2$ and $\chi(t) = \chi(t_1) = \chi(t_2)$. We put $D[t, \texttt{bias}, \texttt{weight}, \texttt{seen}, \texttt{future}] = 1$ if and only if there exist states $(\texttt{bias}_1, \texttt{weight}_1, \texttt{seen}_1, \texttt{future}_1)$ and $(\texttt{bias}_2, \texttt{weight}_2, \texttt{seen}_2, \texttt{future}_2)$ with $D[t_1, \texttt{bias}_1, \texttt{weight}_1, \texttt{seen}_1, \texttt{future}_1] = 1$ and $D[t_2, \texttt{bias}_2, \texttt{weight}_2, \texttt{seen}_2, \texttt{future}_2] = 1$ such that for all $u \in \chi(t)$ and all $x \in \mathcal{X}$:

- *Agreement on interface.* $\texttt{bias}(u) = \texttt{bias}_1(u) = \texttt{bias}_2(u)$ and $\texttt{future}(x, u) = \texttt{future}_1(x, u) = \texttt{future}_2(x, u)$.
- *Agreement of in-bag weights.* For every arc $(a, b) \in E(G)$ with $a, b \in \chi(t)$ we have $\texttt{weight}(a, b) = \texttt{weight}_1(a, b) = \texttt{weight}_2(a, b)$.
- *Additivity of in-subtree contributions.* If we combine a $V_{t_1}$-partial and a $V_{t_2}$-partial neural network, then the seen pre-activation values are disjoint except for values received from neuron in the bag of $t$. Thus, we require that

$$\texttt{seen}(x, u) = \texttt{seen}_1(x, u) + \texttt{seen}_2(x, u) - \sum_{w \in (\chi(t') \cap N^-(u))} \texttt{weight}(w, u) \cdot a_w(x),$$

(note that $\texttt{seen}$ includes activations received from the current bag). As before, $a_w(x)$ is the activation of neuron $w$. Note that, because of the agreement conditions, this value is consistent among the three bags and, as before, can be computed in topological order from $\texttt{bias}$ and the totals $\texttt{seen}$ and $\texttt{future}$.

- *Consistency of out-of-subtree contributions.* The future pre-activation values of a $V_{t_1}$-partial neural network distribute over the pre-activation values seen in $V_{t_2} \setminus V_{t_1}$ and those in $V \setminus (V_{t_1} \cup V_{t_2})$. Analogously for a $V_{t_2}$-partial neural network. Thus, we require:

$$\texttt{future}_1(x, u) + \texttt{seen}_1(x, u) = \texttt{future}_2(x, u) + \texttt{seen}_2(x, u).$$

*Running time.* Let $b := |\chi(t)| \leq 2\mathrm{tw} + 1$ and $p := |\mathcal{X}|$ (recall that $\chi(t)$ is the bag of $t$ and $\mathcal{X}$ is the set of input vectors). A state at $t$ now consists of:

- $\texttt{bias} : \chi(t) \to \mathbb{Z}_d$  ($d^b$ choices),
- $\texttt{weight} : \{(u, v) \in E(G) \mid u, v \in \chi(t)\} \to \mathbb{Z}_d$  ($d^{m_t}$ choices, where $m_t := |E(G[\chi(t)])| \leq b^2$),
- $\texttt{seen}, \texttt{future} : \mathcal{X} \times \chi(t) \to \mathbb{Z}_{d^2 \Delta}$  $(((2d^2 \Delta))^{pb}$ choices each).

Thus the number of table entries per bag is at most

$$d^{b+m_t} \cdot (2d^2 \Delta)^{2pb} \leq d^{b+b^2} \cdot (2d^2 \Delta)^{2pb}.$$

It is not hard to see that each table entry for leaf, introduce, and forget nodes can be computed in polynomial time in $p, b$. Two entries of children of join nodes define an entry of a join node. Thus, the total running time is

$$2^{\mathcal{O}(\mathrm{tw})} \cdot |V(G)| + \mathcal{O}(\mathrm{tw} \cdot |V(G)|) \cdot d^{2b+2b^2} \cdot (2d^2 \Delta)^{4pb} \cdot \mathrm{poly}(pb) = d^{\mathcal{O}(\mathrm{tw} \cdot d^{\mathcal{O}(\alpha)})} \cdot |V(G)|$$

where $\Delta = d^{\mathcal{O}(\alpha d^\alpha)}$. Hence the algorithm runs in time $f(\mathrm{tw}, \alpha, \omega, d) \cdot \mathrm{poly}(|V(G)| + |E(G)|)$, i.e., it is FPT with respect to tw, $\alpha$, and $\omega$. This completes the proof of Lemma 3. $\qquad\square$

Instances with nonzero error bounds can be reduced to the $\ell = 0$ setting in order to apply Lemma 3.

**Theorem 5.** *$d$-QNNT is* FPT *wrt. the treewidth of $G$, the number $\alpha$ of input dimensions, and the number $\omega$ of output dimensions.*

*Proof.* First, in $\mathcal{O}(2^{d^{\alpha+\omega}})$ time we determine (by trying all possibilities) which input-output pairs will not be learned correctly. Note that these can simply be ignored during training. Hence, we may now assume that the error bound $\ell$ is 0 and we need to learn all input-output pairs correctly. Thus, we can apply Lemma 3 to obtain the desired running time. $\qquad\square$

**Theorem 6.** *$d$-QNNT is* FPT *w.r.t. the treewidth of $G$, the number $\alpha$ of input dimensions, and the error bound $\ell$.*

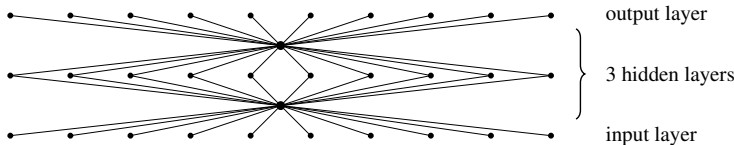

Figure 6: An illustration of an architecture in which the treewidth is significantly smaller than the width. More precisely, $\mathsf{tw} = 2$ and width $= 8$. Moreover, if the second hidden layer were to consist of $p$ neurons, then we would have width $= p$ while preserving $\mathsf{tw} = 2$.

*Proof.* Let $(G, \alpha, \omega, d, \mathcal{D}, \ell)$ be an instance of $d$-QNNT. First, observe that for each input vector $x$ there are at most $\ell + 1$ distinct output vectors as, otherwise, the error bound $\ell$ could not be achieved. Thus, we may first, in $\mathcal{O}(2^{\ell d^\alpha})$ time, determine by trying all possibilities which input-output pairs will not be learned correctly. Note that these can simply be ignored during training. Hence, we may now assume that the error bound $\ell$ is 0 and we need to learn all input-output pairs correctly. Thus, we can apply Lemma 3 to obtain the desired running time. $\square$

For an illustration that there exist architectures in which the treewidth is much smaller than the width, we refer to Figure 6.

**Corollary 1.** *$d$-QNNT is* FPT *with respect to $\alpha + \ell + $ width.*

*Proof.* If there is at least one hidden neuron, by Observation 1, we have that the treewidth is at most two times the width of the architecture. Hence, in this case the result follows from Theorem 6. Otherwise, the architecture is a bipartite graph consisting only of the input and output neurons. The weights of arcs to one output neuron do not influence the activations of other output neurons and hence the problem reduces to solving $\omega$ pairwise independent instances in which there is exactly one output neuron. That is, the original instance is a yes-instance if and only if all of these instances are yes-instances. Each of the single-neuron instances has an architecture of size $\mathcal{O}(\alpha)$ and thus can be solved by brute force in $f(\alpha) \cdot |\mathcal{D}|$ time. Thus, if there are no hidden neurons, we can solve the problem in $f(\alpha) \cdot |\mathcal{D}| \cdot \omega \cdot |V(G)|$ time, as required. $\square$

**Corollary 2.** *$d$-QNNT is* FPT *with respect to $\alpha + \omega + $ width.*

*Proof.* If there is at least one hidden neuron, by Observation 1, we have that the treewidth is at most two times the width of the architecture. Hence, in this case the result follows from Theorem 5. Otherwise, the architecture is a bipartite graph consisting only of the input and output neurons. It thus has size $\mathcal{O}(\alpha \cdot \omega)$ and the corresponding instance can be solved by brute force in $f(\alpha, \omega) \cdot |\mathcal{D}|$ time. $\square$

## 5 CONCLUDING REMARKS

Our work initiates the study of fully quantized ReLU neural network training from the classical as well as parameterized complexity perspectives. We show that the problem remains NP-hard even in highly restricted settings, but also provide positive results through the identification of non-trivial fixed-parameter tractable fragments. We remark that the latter outcome contrasts the state of the art for neural network training in the non-quantized setting. Indeed, in spite of being targeted by several recent complexity-theoretic studies (Dey et al., 2020; Abrahamsen et al., 2021; Goel et al., 2021; Boob et al., 2022; Froese & Hertrich, 2023; Bertschinger et al., 2023; Brand et al., 2023), to date we do not know a single *non-trivial*[4] parameterization that yields fixed-parameter tractability for training non-quantized neural networks. Moreover, we believe that settling the parameterized complexity of $d$-QNNT w.r.t. the input and output dimensionality (i.e., $\alpha + \omega$) will require insights beyond the current state of the art and pose this as the main open question arising from our work. Other important avenues of future work include whether our results can be extended to distillation, and whether they could be used to obtain more efficient empirical algorithms.

---

[4]By non-trivial, we mean that the parameter does not simply bound the input size.

ACKNOWLEDGMENTS

Robert Ganian acknowledges support by the Austrian Science Fund (FWF projects 10.55776/Y1329 and 10.55776/COE12).

Frank Sommer was supported by the Alexander von Humboldt Foundation and partially by the Carl Zeiss Foundation, Germany, within the project "Interactive Inference".