# OpenReview forum: "Tractability via Low Dimensionality: The Parameterized Complexity of Training Quantized Neural Networks"
_ICLR.cc/2026/Conference — ICLR 2026 Poster_

### Official Review · Reviewer_D5xX · 2025-10-31

**Soundness:** 3
**Presentation:** 3
**Contribution:** 3
**Rating:** 6
**Confidence:** 2

**Summary:**

This paper provides a foundational study of the computational and parameterized complexity of quantized neural network training (QNNT). They prove that training remains NP-hard in general but identify tractable cases when parameterized by input dimension combined with width, error bound, or treewidth. Central to their approach is a structural lemma derived from Steinitz’ Lemma, which limits the number of relevant input connections per neuron, enabling dynamic programming–based fixed-parameter tractable algorithms.

**Strengths:**

- This is the first rigorous parameterized complexity analysis of quantized neural network training.

- The authors establish NP-hardness and W[2]-hardness for even extremely constrained architectures (binary weights, single hidden layer), demonstrating the robustness of the hardness landscape.

- It bridges machine learning and complexity theory, setting a precedent for similar analyses in quantized deep learning settings.

**Weaknesses:**

- The connection to practical quantized training algorithms such as quantization-aware training (QAT), could be discussed more deeply. The current presentation is fully theoretical.

**Questions:**

- To what extent can the results in this paper be leveraged for actual algorithmic design in quantized training?

---

> ### Author Response · Authors · 2025-11-20
>
> Thank you for your review!
>
> Indeed, quantization-aware training is one of the methods to obtain a quantized neural network, among others such as fully-quantized training, mixed-precision training, and post-training quantization. They all have their own interesting algorithmic subproblems but they have in common the rough general goal of obtaining a quantized neural network that represents the data well. Our aim is to study the complexity of this general goal. We now mention these techniques at the end of the second paragraph of the introduction.
>
>
> About your Question:
> Our focus is on the fundamental complexity-theoretic properties of the problem, and we do not claim immediate implications for quantized training in real-world scenarios. It is possible that certain ideas from our algorithms can lead to improvements in certain settings (e.g., it is known that small-treewidth graphs and networks occur naturally in a variety of scenarios, see https://arxiv.org/abs/2509.06880 for measurements of treewidth in graphs), but an empirical evaluation is beyond the scope of our contributions. We now mention this as an interesting avenue for future research in the concluding remarks.

---

### Official Review · Reviewer_hBha · 2025-11-01

**Soundness:** 3
**Presentation:** 3
**Contribution:** 4
**Rating:** 8
**Confidence:** 4

**Summary:**

This paper studies theory for quantized neural networks, i.e., whose weights are stricted to be integers.
They show various hardness results for the problem of determining whether there exists weights that achieves performance better than some threshold.
They also show tractability in terms of the tree-width of the underlying graph of a neural network architecture.

**Strengths:**

They consider feed-forward neural net architectures that is not just the fully-connected nets. rather, they allow "sparser" connections.
For many of the hardness results, simple architectures (i.e., low depth) already already lead to hard problems.
Discovery of tree-width and connection to tractability for QNNs is also interesting.

**Weaknesses:**

There isn't a lot of intuition for why small tree-width neural networks. what do those graphs look like? It'd be useful if there is more small scale examples e.g., like the one from Figure 4, but for small tree-width architectures.

Since small tree-width architectures are tractable, it'd be nice to see some experiments on say the two-moon dataset. this will help answer: are small tree-width neural networks useful?

**Questions:**

I don't understand the sentence on Line 468-469. How does the tree width becomes replaced with (ordinary) width?

As ell becomes larger, the problem should become easier. Is this correct? I'm not understanding how ell is used alongside with other parameters like the width which makes the problem harder when the width grows larger.

---

> ### Author Response · Authors · 2025-11-20
>
> Thank you for the encouraging words!
>
> We added an example of a small-treewidth architecture in the revised version (Figure 6 in the full version).
> We agree that experimental evaluations would be an interesting topic for future work and have added a remark about this in the concluding remarks; however, these are beyond the scope of our submission (which focuses on fundamental barriers to tractability rather than empirical aspects).
>
> For Question 1: This sentence refers to the strategy employed in the proofs of Corollaries 1 and 2 (the full proofs of which are in the appendix). Briefly, if there is at least one hidden layer, then the width is an upper bound for the treewidth and Theorems 5 and 6 directly apply also when ‘treewidth’ is replaced by ‘width’. If there is no hidden layer, then we design two simple ad-hoc strategies that learn the neural networks optimally in this case (see the proofs of the Corollaries in the full version for details). The intended meaning is also clarified in the revised version.
>
> For Question 2: Intuitively having a very large ell “should” make the problem easier, and indeed the problem is trivial if ell is large enough. On the other hand, our results show that having a very small ell also makes the problem easier - if the number of errors is constant, then we can reduce the problem to the Exact Training case which is easier. So one interpretation of several of our results is that the hardest cases are when the number of errors is neither too large nor too small.

---

### Official Review · Reviewer_VvcW · 2025-11-04

**Soundness:** 3
**Presentation:** 3
**Contribution:** 2
**Rating:** 6
**Confidence:** 3

**Summary:**

This paper studies the fixed-parameter tractability (FPT) of training quantized neural networks (namely networks with all weights discretized to a certain grid). The basic question is as follows: given a certain neural network architecture (with ReLU activations), discretization level, and dataset, is it feasible to check whether there is a neural network (of that architecture and discretization) achieving low error on that dataset? The fixed-parameter setting is obtained when we "fix" certain parameters (such as depth, width, input dimension, output dimension, etc, and combinations thereof) to be bounded by $k$ and ask whether there is at least an algorithm running in time $f(k) n^{O(1)}$ for some function $f$. As there are many ways to parameterize a neural network, there are many variants of this question. This paper makes significant progress towards answering most natural variants. For example, one of the main results is that the problem is indeed FPT with respect to width + input dimension + output dimension. However, it is _not_ FPT with respect to just width + input dimension. Many other combinations are considered in the paper, and it gives a fairly comprehensive set of results detailing this complexity-theoretic landscape. It is worth noting that the landscape for the real-valued version of this problem (i.e., not quantized) is not quite as rich; the real-valued version is known to be complete for the large class $\exists \mathbb{R}$, whereas the quantized version necessarily lies in NP.

**Strengths:**

Understanding the complexity of training quantized neural networks is a natural and timely theoretical question. The authors have made a commendable effort to map out the FPT landscape of this problem fairly comprehensively. The techniques in the work may be of technical interest to others in this area, especially the use of Steinitz's Lemma to enable some key simplifications. (However, note that I was not able to verify all the proofs carefully.)

**Weaknesses:**

My concern with this paper is that the core problem, while natural in a complexity-theoretic sense, is also quite theoretical and stylized. It is subject to all the usual criticisms of worst-case complexity theory, and as such feels unlikely to have much bearing on training quantized neural networks in practice. This is fine --- I do think the contributions are worthwhile from a complexity-theoretic point of view, and I do understand that such topics have a place in the machine learning literature. I just wonder if a better audience for this particular set of results might be found at say COLT or CCC rather than ICLR.

**Questions:**

1. It would be really useful to have a more detailed discussion of what makes this problem different from two closely related ones: (a) training real-valued networks, and (b) Boolean satisfiability problems (esp circuits or CSPs). It feels like this problem (training quantized neural networks) has flavors of both, and it would be good to understand exactly what techniques do and do not carry over, and where novel ideas are really needed.
2. Probably the most common way in which quantized neural networks are actually obtained is through distillation, i.e. quantizing a higher-precision model. I am curious if the authors have thoughts about this and whether this might make the problem different. For example, consider the problem where the dataset is assumed to be realizable by a network of quantization level poly(d) (or even real-valued), and the question is to check whether there is a network of quantization level d that realizes it. Basically, trying to model the distillation setting.

---

> ### Author Response · Authors · 2025-11-20
>
> Thank you for your review!
> Below you can find our responses to your two questions:
>
> Question 1:
> Compared to (a), the discretization of values in QNNT allows the application of branching techniques and those are used, e.g., in the proofs of Theorems 5 and 6. This is expected, as (a) is ER-complete while QNNT is NP-complete. The Dynamic Programming subroutine employed in the proof of Lemma 3 also relies on discrete values of the variables.
> As for a comparison to (b), in SAT we only seek a single satisfying assignment - but the QNNT problem needs to consider many possible weight assignments and coefficients throughout the network while dealing with a potentially large number of input data points. On an intuitive level, this makes QNNT significantly more challenging: for instance, SAT is trivially FPT when parameterized by the number of variables, but QNNT is NP-hard even when the number of input neurons is 2 (Theorem 4). Moreover, SAT is well-known to be FPT when parameterized by the treewidth of the formula (represented as either a “primal” or “incidence” graph), but QNNT is NP-hard even on architectures with treewidth 1 (implied by Theorem 1).
> The above intuition is informal, and so we have not added it to the submission. However, if you believe it should be there, we can make space and incorporate it (e.g., in the Related Work).
>
> Question 2:
> If we understand correctly, you are asking about whether or to what extent one can translate quality guarantees (i.e., error guarantees) for higher quantization levels to lower quantization levels. That is an interesting question, and we have not thought about this. Our results do not immediately provide an answer, but we now mention it in the updated concluding remarks.

---

### Meta-Review · Area_Chair_hhyw · 2026-01-02

**Summary:**

This paper analyzes the training phase of neural networks, with a particular focus on considering also quantization aspects, which is typically negrected. The authors provide a systematic and very comprehensive complexity-theoretic study of the training of ReLU networks when using quantization, which for instance contains strict lower bounds for the binary setting.

I am not sure what I shall put here about "Provide a summary of the reviewers' concerns that informed your suggested decision for this paper.", when all concerns (except for one, see below) were addressed. The main conerns were:
* too theoretical
* specific concerns about the chosen model
* missing connection to quantized training algorithms

**Reviewer Concerns:**

Review VvcW, second question was "Probably the most common way in which quantized neural networks are actually obtained is through distillation, i.e. quantizing a higher-precision model. I am curious if the authors have thoughts about this and whether this might make the problem different. For example, consider the problem where the dataset is assumed to be realizable by a network of quantization level poly(d) (or even real-valued), and the question is to check whether there is a network of quantization level d that realizes it. Basically, trying to model the distillation setting." and the authors replied "Question 2: If we understand correctly, you are asking about whether or to what extent one can translate quality guarantees (i.e., error guarantees) for higher quantization levels to lower quantization levels. That is an interesting question, and we have not thought about this. Our results do not immediately provide an answer, but we now mention it in the updated concluding remarks.". Hence , this was not fully answered, but it seem to have been out of the scope of the paper, hence I would not rate this negatively.

All other questions and concerns were to my mind answered.

**Reviewer Scores:**

It’s really hard to say how any reviewer would have changed their score if they had taken part more fully in the discussion. Without hearing it from them directly, anything we write here would just be guesswork.

The current scores are 6, 8, and 6. After the to my mind successful rebuttal period, the scores would stay at least the same, or be raised. In any case, it is an accept.

---

### Decision · Program_Chairs · 2026-01-26

Accept (Poster)